# PyTDC: A multimodal machine learning training, evaluation, and inference platform for biomedical foundation models

**Alejandro Velez-Arce** [1 2]   **Marinka Zitnik** [3 4 5 6]

## Abstract

Existing biomedical benchmarks do not provide end-to-end infrastructure for training, evaluation, and inference of models that integrate multimodal biological data and a broad range of machine learning tasks in therapeutics. We present PyTDC, an open-source machine-learning platform providing streamlined training, evaluation, and inference software for multimodal biological AI models. PyTDC unifies distributed, heterogeneous, continuously updated data sources and model weights and standardizes benchmarking and inference endpoints. This paper discusses the components of PyTDC's architecture and, to our knowledge, the first-of-its-kind case study on the introduced single-cell drug-target nomination ML task. We find state-of-the-art methods in graph representation learning and domain-specific methods from graph theory perform poorly on this task. Though we find a context-aware geometric deep learning method that outperforms the evaluated SoTA and domain-specific baseline methods, the model is unable to generalize to unseen cell types or incorporate additional modalities, highlighting PyTDC's capacity to facilitate an exciting avenue of research developing multimodal, context-aware, foundation models for open problems in biomedical AI.

[1]ArcellAI Inc.   [2]calculus.house, The Residency Inc. [3]Department of Biomedical Informatics, Harvard Medical School, Boston, MA, USA [4]Broad Institute of MIT and Harvard, Cambridge, MA, USA [5]Harvard Data Science Initiative, Harvard University, Cambridge, MA, USA [6]Kempner Institute for the Study of Natural and Artificial Intelligence, Harvard University, Cambridge, MA, USA. Correspondence to: Alejandro Velez-Arce <alejandro.velez.arce@@gmail.com>, Marinka Zitnik <marinka@hms.harvard.edu>.

*Proceedings of the 42nd International Conference on Machine Learning*, Vancouver, Canada. PMLR 267, 2025. Copyright 2025 by the author(s).

## 1. Introduction

The application of transfer learning in biology has gained momentum, marking a pivotal shift toward harnessing the wealth of biological measurements available today. Foundation models pre-trained on structural, genomic, and network biology data have demonstrated remarkable potential in improving predictions in data-limited scenarios and accelerating biological discoveries (Theodoris, 2024). These models capture fundamental patterns in biological systems, creating a foundation for tasks ranging from disease modeling to therapeutic development. As this field advances, the need for robust strategies to benchmark these models becomes increasingly important. (Liu et al., 2023) illustrate the importance of diverse downstream task evaluations in assessing the utility of network biology foundation models. However, to be useful in therapeutics, benchmarking must also align with biomedical goals. Focusing benchmarks solely on canonical biology tasks such as cell type clustering and annotation neglects deeper biological insights, such as understanding disease mechanisms. Therefore, developing benchmarks prioritizing therapeutic relevance is essential to guide the evolution of models and ensure they address the challenges at the heart of biomedical research.

Single-cell genomics has enabled the study of cellular processes with remarkable resolution, offering insights into cellular heterogeneity and dynamics (Luecken et al., 2023). Progress in data generation and computational methods designed for single-cell analysis (CZI Single-Cell Biology, et al., 2023) has facilitated context-aware machine learning models that incorporate cell-type-specific data across various therapeutic areas (Chen et al., 2024). Despite these advances, there remains a need for datasets, benchmarks, and tools for reproducible development and benchmarking of out-of-distribution generalization (OOD) (Ektefaie et al., 2024) and domain-specific metric (Velez-Arce et al., 2024) performance of methods that integrate single-cell analysis with diverse therapeutic approaches. Given key problems in genomics intrinsically involve multiple modalities and adapting general-purpose sequence models for these tasks remains unclear (Garau-Luis et al., 2024), any system facilitating method development in single-cell therapeutics must incorporate numerous biomedical modalities.

Kahn et al. (2022) shows us that when domain-specific demands on machine learning solutions increase, framework innovation is often the catalyst for accelerated scientific research by prioritizing open, modular, customizable internals and state-of-the-art, research-ready models and training setups across a variety of domains. As such, we've developed the first (table 1) platform facilitating the curation of multimodal single-cell data, retrieval and inference of context-aware biomedical representation learning models, and therapeutic-task-specific method fine-tuning and benchmarking. We present PyTDC (figure 1), a machine-learning platform providing streamlined training, evaluation, and inference software for single- cell biological foundation models. PyTDC introduces an API-first architecture (Beaulieu et al., 2022) that unifies heterogeneous, continuously updated data sources. The platform introduces a model server (figure 2), which provides unified access to continuously updated model weights across distributed repositories and standardized inference endpoints. Building upon Therapeutic Data Commons (Huang et al., 2021; 2022; Velez-Arce et al., 2024), we present single-cell therapeutics tasks, datasets, and benchmarks for model development and evaluation on OOD predictions and domain-specific metrics.

Overall, the contributions can be summarized as (highlighted in Table 1):

1. **We integrate single-cell analysis with multimodal machine learning in therapeutics via the introduction of contextualized tasks.** PyTDC presents datasets, models, and benchmarks for: single-cell drug-target nomination (Li et al., 2024; Velez-Arce et al., 2024) (section 4, 4.3, and A.1), single-cell chemical and genetic perturbation response predictions (Hetzel et al., 2022; Roohani et al., 2023) (section A.2), and cell-type-specific protein-peptide interaction prediction (Grazioli et al., 2022a; Brown et al., 2023) (section A.3). In our benchmarks we measure: the ability of models perform at most predictive biological contexts (Kwon et al., 2024), perturbation response prediction performance at out-of-distribution samples (Piran et al., 2023), and model robustness to negative sampling heuristics for out-of-distribution predictions (Kim et al., 2023).

2. **We introduce a collection of continually updated heterogeneous biomedical data sources enabled via a novel architecture.** PyTDC's modalities span but are not limited to: single-cell gene expression atlases (CZI Single-Cell Biology, et al., 2023; Jones et al., 2022), single-cell chemical (Aissa et al., 2021; Klein, 2023) and genetic (Norman et al., 2019; Replogle et al., 2022) perturbations (Peidli et al., 2024), clinical trial outcome data (Fu et al., 2022), peptide sequence data (Grazioli et al., 2022b; Gao et al., 2023), peptidomimetics protein-peptide interaction data from AS-MS spec-

troscopy (Brown et al., 2023; Ye et al., 2022), 3D structural protein data (Meslamani et al., 2011; Mysinger et al., 2012; Liu et al., 2015), cell-type-specific protein-protein interaction networks at single-cell resolution (Li et al., 2024), and biomedical knowledge graphs (Chandak et al., 2023). The novel API-first (Beaulieu et al., 2022) (section 3.1) architecture provides abstractions for complex data processing pipelines (Huang et al., 2021), data lineage (Thiago et al., 2020), and versioning (van der Weide et al., 2017). It augments the stability of emerging biomedical AI workflows, based on advancements from (Schick et al., 2023; Patil et al., 2023; M. Bran et al., 2024), with continuous data updates (Liu et al., 2022). More details in section 3.1 and B.1.

3. **We have released open source model retrieval and deployment software that streamlines AI inferencing, downstream fine-tuning, and domain-specific evaluation for representation learning models across biomedical modalities.** The PyTDC model server (sections 3.2 and C and figures 2 and 1) facilitates the use and evaluation of biomedical foundation models for downstream therapeutic tasks and enforces alignment between new method development and therapeutic objectives. We have released support for inference on several state-of-the-art (SoTA) models (Lopez et al., 2018; Theodoris et al., 2023; Cui et al., 2024; Li et al., 2024; ESM Team, 2024) (section C) as well as benchmarking software for domain-specific evaluation (Velez-Arce et al., 2024).

In this work (section 4), we present a case study on single-cell drug-target nomination (Li et al., 2024; Velez-Arce et al., 2024). PyTDC facilitates our analysis showing inadequate performance of domain-specific methods and state-of-the-art (SoTA) general-purpose graph representation learning methods. Using a novel and adaptable framework for contextualized metrics (sections 3.3 and 4.2), we measure therapeutic relevance of these methods by their predictions at most predictive cell types (Zhang et al., 2022; Kwon et al., 2024) and find superior performance by (Li et al., 2024) on these domain-specific metrics. Though a promising result, the inability of (Li et al., 2024) to generalize to unseen cell types and incorporate additional modalities leaves this as an exciting machine learning research task in therapeutics for the PyTDC-enabled development of methods leveraging foundation models and transfer learning.

## 2. Related Work

**Datasets, benchmarks, and platforms.** Numerous datasets, benchmarks, and platforms have been released supporting training and evaluation of predictive models on therapeutic tasks (Wu et al., 2018; Huang et al., 2021; Xu et al., 2022; Huang et al., 2022; Zhu et al., 2022; Li et al., 2022).

*Table 1.* **Assessment of SoTA datasets, benchmarks, and ML platforms** Focusing on single-cell therapeutics, data modalities, and model inferencing support, we compare PyTDC's capabilities, including features developed by the (Velez-Arce et al., 2024) team, with the SoTA platforms (Wu et al., 2018; Huang et al., 2021; Zhu et al., 2022; CZI Single-Cell Biology, et al., 2023; Luecken et al., 2023; Peidli et al., 2024). We find PyTDC outperforms in its coverage of single-cell therapeutic tasks (sections 4 and A) and data modalities (sections 3.1 and B) and in its support of novel inferencing capabilities for representation learning models (sections 3.2 and C) and domain-specific (section 3.3) evaluation of transfer learning methods.

| Feature | TDC | TorchDrug | MoleculeNet | OpenProblems | scPerturb | CELLXGENE | PyTDC |
|---|---|---|---|---|---|---|---|
| Single-cell Drug-Target Identification Task | ✗ | ✗ | ✗ | ✗ | ✗ | ✗ | ✓ |
| CRISPR-Cell Perturbation Response Prediction | ✗ | ✗ | ✗ | ✗ | ✓ | ✗ | ✓ |
| Drug-Cell Perturbation Response Prediction | ✗ | ✗ | ✗ | ✗ | ✓ | ✗ | ✓ |
| Single-cell Protein-Peptide Binding Interaction Prediction | ✗ | ✗ | ✗ | ✗ | ✗ | ✗ | ✓ |
| Structure-Based Drug Design | ✗ | ✓ | ✓ | ✗ | ✗ | ✗ | ✓ |
| Clinical Trial Outcome Prediction | ✗ | ✗ | ✗ | ✗ | ✗ | ✗ | ✓ |
| Knowledge Graphs | ✗ | ✓ | ✗ | ✗ | ✗ | ✗ | ✓ |
| Python API | ✓ | ✓ | ✗ | ✓ | ✗ | ✓ | ✓ |
| Single-cell Gene Expression Atlases | ✗ | ✗ | ✗ | ✓ | ✓ | ✓ | ✓ |
| External API data retrieval | ✗ | ✗ | ✗ | ✗ | ✗ | ✗ | ✓ |
| Data Views | ✗ | ✗ | ✗ | ✗ | ✗ | ✗ | ✓ |
| Pre-computed representation learning model embedding retrieval | ✗ | ✗ | ✗ | ✗ | ✗ | ✓ | ✓ |
| AI Inferencing for pre-trained representation learning models | ✗ | ✓ | ✗ | ✗ | ✗ | ✗ | ✓ |
| AI Inferencing for pre-trained transformer models | ✗ | ✗ | ✗ | ✗ | ✗ | ✗ | ✓ |

There have also been more recent releases of benchmarks for canonical single-cell biology ML tasks (CZI Single-Cell Biology, et al., 2023; Luecken et al., 2023), including of representation learning models in single-cell biology (Initiative, 2023). None of these systems are tailored to meaningful evaluation of representation learning methods (Lopez et al., 2018; Cui et al., 2024; Theodoris et al., 2023; Chen et al., 2024; Li et al., 2024) at therapeutic tasks (Velez-Arce et al., 2024). PyTDC has been developed with the datasets, architecture, and model training, evaluation, and inference capabilities detailed in table 1, figures 2 and 1, and section 3, to address the need for a multimodal machine learning platform for the development of biomedical methods leveraging transfer learning (Theodoris, 2024).

**Therapeutic and single-cell foundation models.** Foundation models trained on Therapeutics Data Commons (Huang et al., 2021; 2022; Velez-Arce et al., 2024) have been shown to generalize across several therapeutic tasks (Zambrano Chaves et al., 2024). Additionally, parallel efforts in training foundation models on large single-cell atlases have shown a potential to advance cell type annotation and matching of healthy-disease cells to study cellular signatures of disease (Yang et al., 2022; Hao et al., 2024; Cui et al., 2024). PyTDC bridges these efforts by providing formal definitions of therapeutic tasks, datasets, and benchmarks at single-cell resolution in order to provide precise therapeutic predictions incorporating cell type contexts (Kwon et al., 2024; Li et al., 2024) and developing machine learning infrastructure for streamlining transfer learning method (Theodoris, 2024) development, evaluation, and inference.

## 3. Streamlining training, evaluation, and inference

### 3.1. Multimodal Single-cell Data Retrieval

A collection of multimodal continually updated heterogeneous data sources is unified under the introduced "API-first-dataset" architecture. Inspired by API-first design (Beaulieu et al., 2022), this microservice architecture is implemented using the model-view-controller design pattern (Bucanek, 2009; Malik et al., 2021) to enable multimodal data views (Churi et al., 2016; Emmerik et al., 1993; Rammerstorfer & Mössenböck, 2003) under a domain-specific-language (Membarth et al., 2016). We discuss further details and data sources for the modalities in section 1 and tasks in table 1.

### 3.2. PyTDC Model Server

PyTDC presents open source model retrieval and deployment software that streamlines AI inferencing and exposes state-of-the-art, research-ready models and training setups for biomedical representation learning models across modalities. In section C and figure 2, we describe the architectural components of the model server and its integration into the model development lifecycle with the TDC platform. In figure 1, we show the components of PyTDC enabling the full development lifecycle for transfer learning ML methods in therapeutics. Figure 3 also shares how a workflow spanning hundreds-to-thousands of lines of code is simplified to below 30 with PyTDC's model server and data retrieval capabilities.

### 3.3. Context-specific metrics

In machine learning applications, data subsets can correspond to critical outcomes. In therapeutics, there is evidence

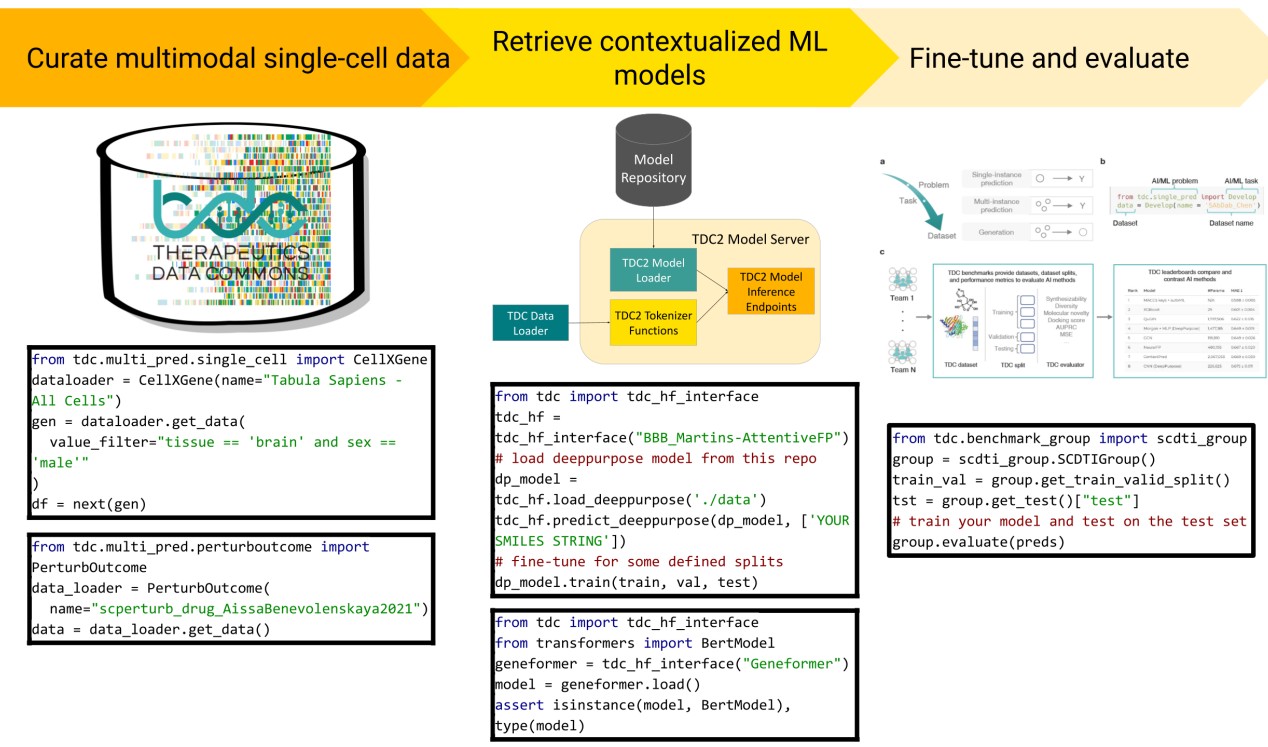

*Figure 1.* We present PyTDC, a machine-learning platform providing streamlined training, evaluation, and inference software for single-cell biological foundation models to accelerate research in transfer learning method development in therapeutics (Theodoris, 2024). PyTDC introduces an API-first (Beaulieu et al., 2022) architecture (sections 3.1 and B) that unifies heterogeneous, continuously updated data sources. The platform introduces a model server, which provides unified access to model weights across distributed repositories and standardized inference endpoints (sections 3.2, B.3.3, and C). The model server accelerates research workflows (Kahn et al., 2022) by exposing state-of-the-art, research-ready models and training setups for biomedical representation learning models across modalities. Building upon Therapeutic Data Commons (Huang et al., 2021; 2022; Velez-Arce et al., 2024), we present single-cell therapeutics tasks, datasets, and benchmarks for model development and evaluation (sections 4 and A).

that the effects of drugs can vary depending on the type of cell they are targeting and where specific proteins are acting (Zhang et al., 2022). Inspired by the success of candidate Graph Foundation Model architectures (Huang et al., 2023) at subgraph-based views for classification tasks (Mao et al., 2024) and building upon the concept of 'slices' (Chen et al., 2020a), we assess model performance on critical biological subsets by introducing the framework and definitions for contextualized metrics to measure model performance at critical biological slices. Our benchmark for single-cell drug-target nomination measures model performance at cell-type-specific subgraphs by computing classification and retrieval metrics aggregated over the top-performing cell types. See Section 4.2 for definitions.

## 4. Evaluation of graph representation learning models at context-aware drug-target nomination

Identifying disease pathways, which are groups of proteins linked to specific diseases, is a crucial area of research that can lead to valuable insights for diagnosing, prognosis, and treating diseases. Computational methods facilitate this discovery process by utilizing protein-protein interaction (PPI) networks. These methods typically begin with a few known disease-associated proteins and seek to identify additional proteins in the pathway by exploring the PPI network surrounding these known proteins. However, these approaches have often had limited success, and the reasons for their failures are poorly understood (Agrawal et al., 2018). Single-cell analysis provides a promising pathway to address this

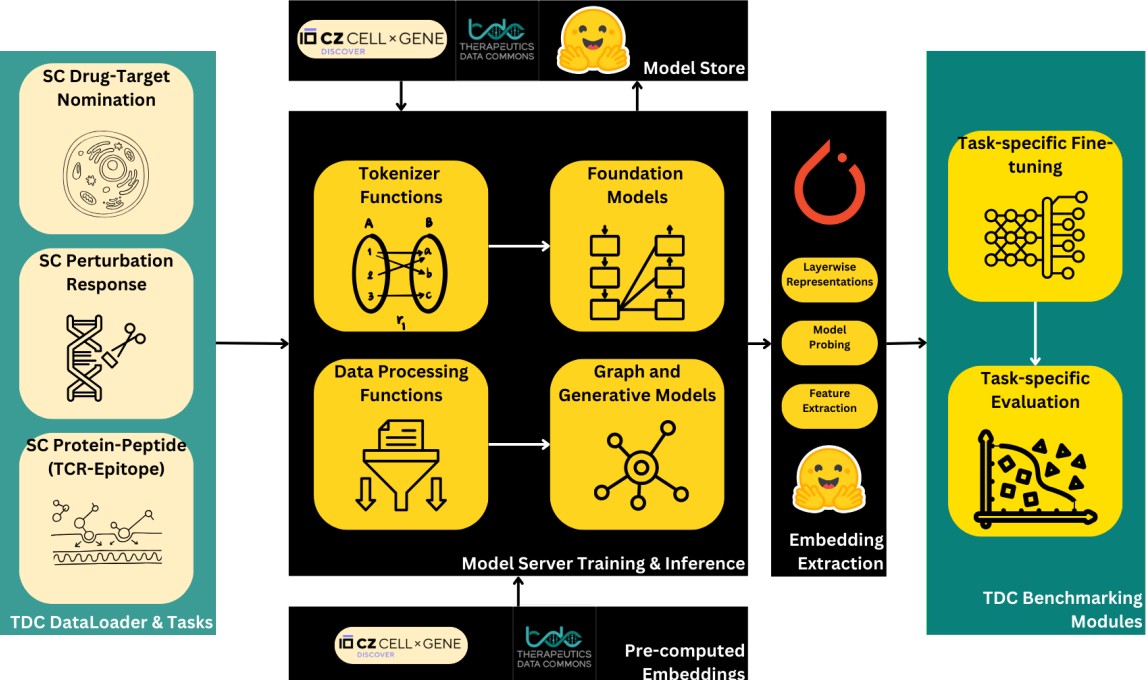

*Figure 2.* **AI inferencing and model evaluation components.** The PyTDC model server (sections 3.2 and C) streamlines retrieval, inferencing, and training setup for an array of context-aware biological foundation models and models spanning multiple modalities. A model store retrieval API provides unified access to model weights stored in the Hugging Face Model Hub (https://huggingface.co/apliko), Chan-Zuckerberg CELLxGENE Census fine-tuned models, and TDC (Huang et al., 2021; 2022; Velez-Arce et al., 2024) storage. The model server also provides access to model classes, tokenizer functions, and inference endpoints supporting PyTorch (Paszke et al., 2019) and Hugging Face Transformers (Wolf et al., 2020). Extracted embeddings, from either model server inference or pre-computed embedding storage, are ready for downstream use by task-specific benchmarking modules.

by enabling the study of gene expression and function at the level of individual cells across healthy and disease states (Jones et al., 2022; CZI Single-Cell Biology, et al., 2023; Kwon et al., 2024). To facilitate biological discoveries using single-cell data, machine-learning models have been developed to capture the complex, cell-type-specific behavior of genes (Lopez et al., 2018; Yang et al., 2022; Theodoris et al., 2023; Cui et al., 2024; Li et al., 2024). PyTDC introduces a task, dataset, and benchmark for the development of contextual AI models to nominate therapeutic targets in a cell type-specific manner (Li et al., 2024).

**Machine Learning Challenges.** Fragmentation of disease and drug pathways when overlaid on networks has been studied quite extensively. In fact, an overwhelming majority of pathways don't correspond to well-connected components in PPI networks, leading to poor performance of state-of-the-art disease pathway discovery methods (Agrawal et al., 2018). One promising technique in machine learning for omics is identifying the most predictive cell-type contexts for a biomarker (Velez-Arce et al., 2024). This approach can help determine the cell types that play crucial and dis-

tinct roles in the disease pathogenesis of conditions like rheumatoid arthritis (RA) and inflammatory bowel diseases (IBD) (Li et al., 2024). To our knowledge, no work systematically evaluates state-of-the-art methods' ability to identify most predictive cell types in drug target nomination.

### 4.1. Single-cell drug-target nomination formulation

Single-cell drug-target nomination (tdc_ml.scDTN) is a predictive, non-generative task formalized as learning an estimator for a disease-specific function $f$ of a target protein and cell type outputting whether the candidate protein $t$ is a therapeutic target in that cell type $c$:

$$y = f(t, c). \tag{1}$$

**Target candidate set.** The target candidate set includes proteins, nucleic acids, or other molecules drugs can interact with, producing a therapeutic effect or causing a biological response. The target candidate set is constrained to proteins relevant to the disease being treated. It is denoted by:

$$\mathbb{T} = \{t_1, \ldots, t_{N_t}\}, \tag{2}$$

*Figure 3.* The below code illustrates the integration of PyTDC's single-cell datasets with the model server (sections 3.2, C, and B.3.3). Here we retrieve the (Peidli et al., 2024; Aissa et al., 2021) chemical perturbation response dataset (section A.2.4, A.2, and A.2.2) and extract gene expression vector embeddings from (Theodoris et al., 2023). Such a workflow would take a user hundreds-to-thousands of lines of code to develop. PyTDC allows the user to extract single-cell foundation model embeddings from complex and customized gene expression datasets with less than 30 lines of code.

```python
from tdc_ml.multi_pred.perturboutcome import PerturbOutcome
from tdc_ml import tdc_hf_interface
import torch

dataset = "scperturb_drug_AissaBenevolenskaya2021"
data = PerturbOutcome(dataset) # import dataset for chemical perturbation response
    prediction
adata = data.adata
tokenizer = GeneformerTokenizer(max_input_size=3) # initialize model tokenizer
adata.var["feature_id"] = adata.var.index.map(
    lambda x: tokenizer.gene_name_id_dict.get(x, 0)) # format features using tokenizer
        data processing functions
x = tokenizer.tokenize_cell_vectors(adata, ensembl_id="feature_id", ncounts="counts") #
    tokenize custom dataset
cells, _ = x
geneformer = tdc_hf_interface("Geneformer")
model = geneformer.load() # load the Geneformer model
input_tensor = torch.tensor(cells)
# note you'd typically batch the input tensor
attention_mask = torch.tensor([[t != 0 for t in cell] for cell in batch])
model(batch,
    attention_mask=attention_mask,
    output_hidden_states=True)
```

where $t_1, \ldots, t_{N_t}$ are $N_t$ target candidates for the drugs treating the disease. Information modeled for target candidates can include interaction, structural, and sequence information.

**Biological context set.** The biological context set includes the cell-type-specific contexts in which the target candidate set operates. This set is denoted as:

$$\mathbb{C} = \{c_1, \ldots, c_{N_c}\}, \tag{3}$$

where $c_1, \ldots, c_{N_c}$ are $N_c$ biological contexts on which drug-target interactions are being evaluated. Information modeled for cell-type-specific biological contexts can include gene expression and tissue hierarchy. The set is constrained to disease-specific cell types and tissues.

**Drug-target nomination.** Drug-Target Nomination is a binary label $y \in \{1, 0\}$, where $y = 1$ indicates the protein is a candidate therapeutic target. At the same time, 0 means the protein is not such a target.

The goal is to train a model $f_\theta$ for predicting the probability $\hat{y} \in [0, 1]$ that a protein is a candidate therapeutic target in a specific cell type. The model learns an estimator for a disease-specific function of a protein target $t \in \mathbb{T}$ and a cell-type-specific biological context $c \in \mathbb{C}$ as input, and the model is tasked to predict:

$$\hat{y} = f_\theta(t \in \mathbb{T}, c \in \mathbb{C}). \tag{4}$$

### 4.2. Context-Specific Metrics in `tdc_ml.scDTN`

Context-specific metrics (see section 3.3) are defined to measure model performance at critical biological slices, with our benchmarks focused on measuring cell-type-specific model performance. For single-cell drug-target nomination, we measure model performance at top-performing cell types. We evaluate predictive cell type retrieval using average precision at rank R (AP@R) and target classification using AUROC, both at top K predicted cell types. Formally, we define context-specific AP@R and AUROC below.

**Context-specific AUROC.** To calculate the AUROC for the top K performing cell types, we first need to determine which cell types achieve the highest AUROC scores. After selecting the top-performing cell types, we weigh each top-performing cell type's AUROC score by the number of samples in that cell type.

We denote:

$$\mathbb{D} = \{(x_i, y_i, c_i)\}, \quad \forall i \in \mathbb{S} \qquad (5)$$

Here, $\mathbb{D}$ denotes the dataset where $x_i$ denotes the feature vector, $y_i$ is the true label, and $c_i$ is the cell type for sample i from $\mathbb{S}$. We further denote $C$, the set of unique cell types. Then, the AUROC for a specific cell type, $AUROC_c$, is computed as:

$$AUROC_c = AUROC(D_c) \qquad (6)$$

Here, $D_c = \{(x_i, y_i)|c_i = c\}$ is the subset of the dataset for cell type c and $AUROC(D_c)$ represents the AUROC score computed over this subset. Once these are computed, values can be sorted in descending order to select the top X cell type with highest AUROC value.

$$C_K = \{c_1, c_2, \ldots, c_K\} \quad \text{s.t.}$$
$$AUROC_{c_i} \geq AUROC_{c_j}, \qquad (7)$$
$$\forall i \leq K, j > K$$

The weighted AUROC for the top K cell types is given by weighting each cell type's AUROC by the proportion of its samples relative to the total samples in the top K cell types.

$$AUROC_{TopK} = \frac{\sum_{c \in C_K} AUROC_c \times |D_c|}{\sum_{c \in C_K} |D_c|} \qquad (8)$$

This measure represents a balance between representation and performance of the cell types, and primarily evaluates the model's classification strengths (Velez-Arce et al., 2024).

**Context-specific Average Precision at rank R (AP@R).** In our study, we compute AP@R for the top K performing cell types. We denote dataset and samples as above.

$$\mathbb{D} = \{(x_i, y_i, c_i)\}, \quad \forall i \in \mathbb{S} \qquad (9)$$

Here, $\mathbb{D}$ denotes the dataset where $x_i$ denotes the feature vector, $y_i$ is the true label, and $c_i$ is the cell type for sample i from $\mathbb{S}$. We further denote $C$, the set of unique cell types. The samples of each cell type, $D_c = (x_i, y_i)|c_i = c$, can be sorted based on the score output by the model for said sample $f(x_i)$, with average precision at rank type computed accordingly.

$$D_c^R = \{x_1, \ldots, x_R\} \quad \text{s.t.}$$
$$f(x_i) \geq f(x_j), \qquad (10)$$
$$\forall i \leq R, j > R, c_i = c, c_j = c$$

$$AP@R_c = AP(\{y_1, \ldots, y_R\}, \{f(x_1), \ldots, f(x_R)\}),$$
$$x_i \in D_c^R$$
$$(11)$$

The average precision at rank R at Top K cell types can then be defined as:

$$C_K = \{c_1, c_2, \ldots, c_K\} \quad \text{s.t.}$$
$$AP@R_{c_i} \geq AP@5_{c_j}, \qquad (12)$$
$$\forall i \leq K, j > K$$

$$AP@R_{TopK} = mean(\{AP@R_{c_i}\}, \quad \forall c_i \in C_K) \qquad (13)$$

AP@R is robust to varied sizes among cell-type-specific subgraphs and number of protein targets in those subgraphs (Li et al., 2024; Velez-Arce et al., 2024) and it is used as the retrieval evaluation metric.

### 4.3. Drug-target Nomination Benchmarks

We use curated therapeutic target labels from the Open Targets Platform (Targets, 2023) for rheumatoid arthritis (RA) and inflammatory bowel disease (IBD) (Li et al., 2024) (section A.1.2). Previous work (Velez-Arce et al., 2024; Li et al., 2024) benchmarked PINNACLE (Li et al., 2024)—trained on cell type specific protein-protein interaction networks—and a graph attention neural network (GAT) (Velickovic et al., 2017)—trained on a context-free reference protein-protein interaction network—on the curated therapeutic targets dataset. The results (Appendix tables 4 and 3) showed PINNACLE outperforming GAT when it's performance on contextualized metrics (section 4.2) was compared to GAT's aggregate, context-free, performance. However, this evaluation does not establish the difficulty of performing well on contextualized metrics for drug-target nomination or motivate the need for stronger machine learning models for this task. To our knowledge, PyTDC is the first to make this case, evaluate state-of-the-art methods at the task, and streamline training, evaluation, and inference in single-cell drug-target nomination.

As a domain-specific baseline, we run a label propagation algorithm Aridhi et al. (2017) over the Li et al. (2024) cell type specific PPI by labeling the train and validation sets and propagating to the unlabeled test set via a majority vote among the neighborhood of an unlabeled node. We find a significant portion of the graph remains unlabeled after convergence, resulting in poor performance of the domain-specific baseline illustrated in table 2. We attribute this result to the fragmentation of RA and IBD pathways when overlaid on PPI networks (Agrawal et al., 2018).

For comparison, we evaluate a state-of-the-art architecture for representation learning with graphs, Brody et al. (2022), which incorporates an attention function. In addition, we evaluate an attention-free architecture for representation learning with graphs, Grover & Leskovec (2016). In both cases, we one-hot encode cell type labels to the context-free PPI graph used to evaluate GAT in Velez-Arce et al. (2024); Li et al. (2024).

*Table 2.* **Cell-type specific target nomination for two therapeutic areas, rheumatoid arthritis (RA) and inflammatory bowel diseases (IBD).** Cell-type specific context metrics (see 3.3 and definitions in Section 4.2) are used to benchmark (Grover & Leskovec, 2016; Aridhi et al., 2017; Brody et al., 2022; Li et al., 2024). Average precision for ranks 5 and 20 for top 20 and 50 cell types is used. AUROC at top 1, 10, 20, and 50 cell types is used.

| Model | AP@5 Top-20 CT | AP@20 Top-20 CT | AP@5 Top-50 CT | AP@20 Top-50 CT |
|---|---|---|---|---|
| PINNACLE (RA) | $0.917_{\pm0.057}$ | $0.625_{\pm0.041}$ | $0.628_{\pm0.057}$ | $0.452_{\pm0.033}$ |
| GATv2 (RA) | $0.334_{\pm0.050}$ | $0.246_{\pm0.055}$ | $0.252_{\pm0.021}$ | $0.187_{\pm0.033}$ |
| Node2Vec (RA) | $0.585_{\pm0.045}$ | $0.448_{\pm0.023}$ | $0.476_{\pm0.028}$ | $0.383_{\pm0.020}$ |
| Label Propagation (RA) | $0.105_{\pm0.024}$ | $0.077_{\pm0.005}$ | $0.042_{\pm0.010}$ | $0.048_{\pm0.005}$ |
| PINNACLE (IBD) | $0.884_{\pm0.071}$ | $0.634_{\pm0.048}$ | $0.582_{\pm0.047}$ | $0.439_{\pm0.030}$ |
| GATv2 (IBD) | $0.348_{\pm0.043}$ | $0.261_{\pm0.007}$ | $0.259_{\pm0.017}$ | $0.189_{\pm0.004}$ |
| Node2Vec (IBD) | $0.617_{\pm0.081}$ | $0.417_{\pm0.044}$ | $0.406_{\pm0.070}$ | $0.295_{\pm0.036}$ |
| Label Propagation (IBD) | $0.268_{\pm0.014}$ | $0.228_{\pm0.005}$ | $0.126_{\pm0.011}$ | $0.134_{\pm0.003}$ |
| **Model** | **AUROC Top-1 CT** | **AUROC Top-10 CT** | **AUROC Top-20 CT** | **AUROC Top-50 CT** |
| PINNACLE (RA) | $0.765_{\pm0.054}$ | $0.676_{\pm0.017}$ | $0.652_{\pm0.013}$ | $0.604_{\pm0.007}$ |
| GATv2 (RA) | 0.5 | 0.5 | 0.5 | 0.5 |
| Node2Vec (RA) | $0.766_{\pm0.030}$ | $0.731_{\pm0.024}$ | $0.715_{\pm0.025}$ | $0.687_{\pm0.024}$ |
| Label Propagation (RA) | $0.451_{\pm0.003}$ | $0.415_{\pm0.002}$ | $0.392_{\pm0.002}$ | $0.364_{\pm0.002}$ |
| PINNACLE (IBD) | $0.935_{\pm0.067}$ | $0.806_{\pm0.023}$ | $0.757_{\pm0.017}$ | $0.672_{\pm0.015}$ |
| GATv2 (IBD) | 0.5 | 0.5 | 0.5 | 0.5 |
| Node2Vec (IBD) | $0.883_{\pm0.116}$ | $0.703_{\pm0.036}$ | $0.666_{\pm0.030}$ | $0.610_{\pm0.031}$ |
| Label Propagation (IBD) | $0.683_{\pm0.000}$ | $0.544_{\pm0.000}$ | $0.511_{\pm0.001}$ | $0.457_{\pm0.000}$ |

Our results (table 2) show Brody et al. (2022); Grover & Leskovec (2016) extract non-random, informative signals by their outperforming label propagation at retrieval, but both methods are far from optimal. SoTA general ML models and domain-specific methods performed poorly at this study's single-cell drug target nomination task, supporting the need for stronger models. Li et al. (2024), which outperformed all methods, is a first step in this direction. Still, it is specifically engineered for cell type-specific PPI networks and can only work with apriori-defined and user-specific contexts. In addition, Li et al. (2024)'s contextualized AUROC was strongest at IBD, where label propagation also performed well (better than (Brody et al., 2022)), but weaker and even inferior to Grover & Leskovec (2016) in RA, where label propagation performed worse than random. This suggests ample room for improvement at generalization out of subgraph-based view Mao et al. (2024), and a promising avenue of research in this task for GFMs and other representation-learning-based transfer learning methods.

## 5. Conclusion

PyTDC represents a significant step in accelearting scientific research by developing machine learning infrastructure (Kahn et al., 2022) for biomedical AI, particularly in single-cell therapeutics (Theodoris, 2024). By integrating multimodal data retrieval (section 3.1), standardized benchmarking (sections 4 and A), and model inference (sections 3.2 and C) into a unified platform, PyTDC enables seamless access to heterogeneous biological datasets and provides a foundation for training and evaluating machine learning models in therapeutics. Introducing an API-first architecture (Beaulieu et al., 2022) facilitates continuous data updates, ensuring that biomedical AI models remain adaptable to discoveries. Moreover, PyTDC's model server (figures 2 and 1) streamlines biomedical foundation models' retrieval, deployment, and fine-tuning, aligning new method development with real-world therapeutic objectives. Our case study on single-cell drug-target nomination highlights key challenges in model generalization and domain-specific evaluation. Despite advances in graph representation learning and context-aware geometric deep learning, existing methods struggle to generalize across unseen cell types and integrate additional biomedical modalities. The introduction of context-specific evaluation metrics within PyTDC provides a quantitative framework for assessing model performance in the most predictive biological contexts, addressing critical gaps in therapeutic model benchmarking. These results emphasize the need for multimodal, context-aware foundation models, a research direction that PyTDC is uniquely positioned to support. By defining open challenges, establishing state-of-the-art baselines, and providing tools for method development and evaluation, PyTDC fosters a more rigorous, reproducible, and therapeutically relevant machine learning ecosystem. PyTDC will accelerate innovation in single-cell AI, drive improvements in domain-specific generalization, and establish new standards for biomedical machine learning research. We invite the community to contribute to this open-source effort and leverage PyTDC as a scalable, adaptable, and transparent framework for advancing therapeutics

through AI.

## Software and Data

PyTDC is open-source at https://github.com/apliko-xyz/PyTDC and documentation is provided via https://pytdc.apliko.io. In addition, notebooks used to perform the benchmarking discussed in this paper can be found at https://github.com/amva13/pinnacle_benchmark.

## Acknowledgements

We thank Jesus Caraballo Anaya, a member of the PyTDC team, for his contributions to the technologies discussed in this manuscript. We thank the TDC team for collecting useful datasets.

We gratefully acknowledge the support of NIH R01-HD108794, NSF CAREER 2339524, US DoD FA8702-15-D-0001, awards from Harvard Data Science Initiative, Amazon Faculty Research, Google Research Scholar Program, AstraZeneca Research, Roche Alliance with Distinguished Scientists, Sanofi iDEA-iTECH Award, Pfizer Research, Chan Zuckerberg Initiative, John and Virginia Kaneb Fellowship award at Harvard Medical School, Aligning Science Across Parkinson's (ASAP) Initiative, Biswas Computational Biology Initiative in partnership with the Milken Institute, Harvard Medical School Dean's Innovation Awards for the Use of Artificial Intelligence, and Kempner Institute for the Study of Natural and Artificial Intelligence at Harvard University. Any opinions, findings, conclusions or recommendations expressed in this material are those of the authors and do not necessarily reflect the views of the funders.

## Impact Statement

**Limitations and societal considerations.** PyTDC software is open-source and available under MIT license. However, models and datasets exposed by PyTDC are released under their own licenses, which though exposed, PyTDC is unable to strictly enforce. It's been the case such licenses have been violated by commercial entities developing models on released datasets (Zambrano Chaves et al., 2024). Open-source datasets and benchmark can increase risk of biased or incomplete data, which may lead to inaccurate or non-representative AI models. Additionally, the open accessibility of such datasets could lead to misuse, including unethical applications or the proliferation of AI models that reinforce existing biases in medicine. Moreover, reliance on standardized benchmarks may discourage innovation and lead to the over-fitting of models to specific datasets, potentially limiting their generalization in real-world scenarios. Last, evaluating deep learning models for genetic perturbation tasks requires reconsideration in light of recent findings that question their effectiveness for this problem. A recent study revealed that deep learning models do not consistently outperform simpler linear models across various benchmarks (Ahlmann-Eltze et al., 2024). PyTDC datasets, benchmarks, metrics, and foundation model tooling lay the foundation for more thorough study, development, and evaluation of models in this space.

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

# A. New tasks: datasets, benchmarks, and code

PyTDC (Velez-Arce et al., 2024) introduces four tasks with fine-grained biological contexts: contextualized drug-target identification, single-cell chemical/genetic perturbation response prediction, and cell-type-specific protein-peptide binding interaction prediction, which introduce antigen-processing-pathway-specific, cell-type-specific, peptide-specific, and patient-specific biological contexts. Benchmarks for drug target nomination, genetic perturbation response prediction and chemical perturbation response prediction, all at single-cell resolution, were computed with corresponding leaderboards introduced on the TDC website. In addition, a benchmark and leaderboard was introduced for the TCR-Epitope binding interaction task for peptide design at single-cell resolution for T-cell receptors.

**Context-specific metrics (section 3.3 and 4.2).** In real-world machine learning applications, data subsets can correspond to critical outcomes. In therapeutics, there is evidence that the effects of drugs can vary depending on the type of cell they are targeting and where specific proteins are acting (Zhang et al., 2022). We build on the "slice" abstraction (Chen et al., 2020a) to measure model performance at critical biological subsets. Context-specific metrics are defined to measure model performance at critical biological slices, with our benchmarks focused on measuring cell-type-specific model performance. In the case of benchmarks for perturbation response prediction and protein-peptide binding affinity, the studies are limited to a particular cell line, however our definition for context-specific metrics lays the foundations for building models which can generalize across cell lines and make context-aware predictions (Kwon et al., 2024). For single-cell drug-target nomination, we measure model performance at top-performing cell types. See corresponding sections 3.3 and 4.2 in the main text.

## A.1. `tdc_ml.scDTI`: Contextualized Drug-Target Identification

**Motivation.** Single-cell data have enabled the study of gene expression and function at the level of individual cells across healthy and disease states (Jones et al., 2022; CZI Single-Cell Biology, et al., 2023; Kwon et al., 2024). To facilitate biological discoveries using single-cell data, machine-learning models have been developed to capture the complex, cell-type-specific behavior of genes (Theodoris et al., 2023; Cui et al., 2024; Yang et al., 2022; Li et al., 2024). In addition to providing the single-cell measurements and foundation models, PyTDC supports the development of contextual AI models to nominate therapeutic targets in a cell type-specific manner (Li et al., 2024). We introduce a benchmark dataset, model, and leaderboard for context-specific therapeutic target prioritization, encouraging the innovation of model architectures (e.g., to incorporate new modalities, such as protein structure and sequences (Li et al., 2020; 2019; Lee et al., 2018; Öztürk et al., 2018; Meng et al., 2017), genetic perturbation data (Ji et al., 2020; Ghadie et al., 2017; Zeng et al., 2019; Wang et al., 2021a), disease-specific single-cell atlases (Carlberg et al., 2019; Kuenzi et al., 2020; Julkunen et al., 2020), and protein networks (Parca et al., 2019; Zhang et al., 2023; Gupta et al., 2022)). PyTDC's release of `tdc_ml.scDTI` is a step in standardizing benchmarks for more comprehensive assessments of context-specific model performance.

### A.1.1. DATASET AND BENCHMARK

We use curated therapeutic target labels from the Open Targets Platform (Targets, 2023) for rheumatoid arthritis (RA) and inflammatory bowel disease (IBD) (Li et al., 2024) (section A.1.2). We benchmark PINNACLE (Li et al., 2024)—trained on cell type specific protein-protein interaction networks—and a graph attention neural network (GAT) (Velickovic et al., 2017)—trained on a context-free reference protein-protein interaction network—on the curated therapeutic targets dataset. As expected, PINNACLE underperforms when evaluated on context-agnostic metrics (Table 3) and drastically outperforms GAT when evaluated on context-specific metrics (Appendix Table 4).

To our knowledge, PyTDC provides the first benchmark for context-specific learning (Kwon et al., 2024). PyTDC's contribution helps standardize the evaluation of single-cell ML models for drug target identification and other single-cell tasks (Yang et al., 2022; Theodoris et al., 2023; Li et al., 2024; Cui et al., 2024).

| Model | AP@5 | AUROC |
|---|---|---|
| PINNACLE (RA) | 0.23 | 0.55 |
| GAT (RA) | 0.22 | 0.58 |
| PINNACLE (IBD) | 0.20 | 0.58 |
| GAT (IBD) | 0.20 | 0.64 |

*Table 3.* RA and IBD results for PINNACLE and a Graph Attention Network without cell-type-specific context metrics. These are results obtained after evaluating 10 seeds.

*Table 4.* **Cell-type specific target nomination for two therapeutic areas, rheumatoid arthritis (RA) and inflammatory bowel diseases (IBD).** Cell-type specific context metrics (definitions in Section 4.2): AP@5 Top-20 CT - average precision at $k = 5$ for the 20 best-performing cell types (CT); AUROC Top-1 CT - AUROC for top-performing cell type; AUROC Top-10 CT and AUROC Top-20 CT - weighted average AUROC for top-10 and top-20 performing cell types, respectively, each weighted by the number of samples in each cell type; AP@5/AUROC CF - context-free AP@5/AUROC integrated across all cell types. Shown are results from models run on ten independent seeds. N/A - not applicable.

| Model | AP@5 Top-20 CT | AUROC Top-1 CT | AUROC Top-10 CT | AUROC Top-20 CT | AP@5 CF | AUROC CF |
|---|---|---|---|---|---|---|
| PINNACLE (RA) | $0.913_{\pm 0.059}$ | $0.765_{\pm 0.054}$ | $0.676_{\pm 0.017}$ | $0.647_{\pm 0.014}$ | $0.226_{\pm 0.023}$ | $0.510_{\pm 0.005}$ |
| GAT (RA) | N/A | N/A | N/A | N/A | $0.220_{\pm 0.013}$ | $0.580_{\pm 0.010}$ |
| PINNACLE (IBD) | $0.873_{\pm 0.069}$ | $0.935_{\pm 0.067}$ | $0.799_{\pm 0.017}$ | $0.752_{\pm 0.011}$ | $0.198_{\pm 0.013}$ | $0.500_{\pm 0.010}$ |
| GAT (IBD) | N/A | N/A | N/A | N/A | $0.200_{\pm 0.023}$ | $0.640_{\pm 0.017}$ |

### A.1.2. (LI, MICHELLE, ET AL.) DATASET

To curate target information for a therapeutic area, we examine the drugs indicated for the therapeutic area of interest and its descendants. The two therapeutic areas examined are rheumatoid arthritis (RA) and inflammatory bowel disease. For rheumatoid arthritis, we collected therapeutic data (i.e., targets of drugs indicated for the therapeutic area) from OpenTargets for rheumatoid arthritis (EFO 0000685), ankylosing spondylitis (EFO 0003898), and psoriatic arthritis (EFO 0003778). For inflammatory bowel disease, we collected therapeutic data for ulcerative colitis (EFO 0000729), collagenous colitis (EFO 1001293), colitis (EFO 0003872), proctitis (EFO 0005628), Crohn's colitis (EFO 0005622), lymphocytic colitis (EFO 1001294), Crohn's disease (EFO 0000384), microscopic colitis (EFO 1001295), inflammatory bowel disease (EFO 0003767), appendicitis (EFO 0007149), ulcerative proctosigmoiditis (EFO 1001223), and small bowel Crohn's disease (EFO 0005629).

We define positive examples (i.e., where the label y = 1) as proteins targeted by drugs that have at least completed phase 2 of clinical trials for treating a specific therapeutic area. As such, a protein is a promising candidate if a compound that targets the protein is safe for humans and effective for treating the disease. We retain positive training examples activated in at least one cell type-specific protein interaction network.

We define negative examples (i.e., where the label y = 0) as druggable proteins that do not have any known association with the therapeutic area of interest according to Open Targets. A protein is deemed druggable if targeted by at least one existing drug. We extract drugs and their nominal targets from Drugbank. We retain negative training examples activated in at least one cell type-specific protein interaction network.

**Dataset statistics.** The final number of positive (negative) samples for RA and IBD were 152 (1,465) and 114 (1,377), respectively. In (Li et al., 2024), this dataset was augmented to include 156 cell types.

**Dataset split. Cold Split**: We split the dataset such that about 80% of the proteins are in the training set, about 10% of the proteins are in the validation set, and about 10% of the proteins are in the test set. The data splits are consistent for each cell type context to avoid data leakage.

**References.** (Li et al., 2024)

**Dataset license.** CC BY 4.0

**Code Sample.** The dataset and splits are currently available on TDC Harvard Dataverse. In addition, you may obtain the protein splits used in (Li et al., 2024) via the following code.

```
from tdc_ml.resource.dataloader import DataLoader
data = DataLoader(name=opentargets_dti)
splits = data.get_split()
```

### A.1.3. CODE FOR DATASETS AND BENCHMARK

Retrieving training and validation datasets for the tdc_ml.scDTN task is as simple as 3 lines of code, with an additional 2 lines enabling model evaluation. Furthermore, PyTDC exposes the PPI network and cell types used for benchmarking of (Grover & Leskovec, 2016; Velickovic et al., 2017; Li et al., 2024; Brody et al., 2022) via a handful of lines of code as well.

**Listing 1** The below code illustrates how to retrieve the train, test, and validation splits and run model evaluation for the tdc_ml.scDTN benchmark.

```
from tdc_ml.benchmark_group import scdti_group

group = scdtn_group.SCDTIGroup()
train, val = group.get_train_valid_split(seed=seed)   # Train model
test = group.get_test()["test"]   # Obtain predictions on the test set
group.evaluate(preds)   # Get benchmark results
```

**Listing 2** The below code illustrates how to retrieve the PPI network and cell type labels (Li et al., 2024) used for training and benchmarking of models in tdc_ml.scDTN. Retrieval of cell type labels and embeddings from (Li et al., 2024) are also supported.

```
from tdc_ml.resource.pinnacle import PINNACLE
graph = pinnacle.get_ppi() # ppi network
embeds = pinnacle.get_embeds() # full pre-computed embeddings dataset
scproteins = pinnacle.get_keys() # protein cell types
```

### A.1.4. `TDC_ML.SCDTI`: CONTEXTUALIZED DRUG-TARGET NOMINATION (IDENTIFICATION)

PyTDC introduces `tdc_ml.scDTI` task. The predictive, non-generative task is formalized as learning an estimator for a disease-specific function $f$ of a target protein and cell type outputting whether the candidate protein $t$ is a therapeutic target in that cell type $c$:

$$y = f(t, c). \tag{14}$$

**Target candidate set.** The target candidate set includes proteins, nucleic acids, or other molecules drugs can interact with, producing a therapeutic effect or causing a biological response. The target candidate set is constrained to proteins relevant to the disease being treated. It is denoted by:

$$\mathbb{T} = \{t_1, \ldots, t_{N_t}\}, \tag{15}$$

where $t_1, \ldots, t_{N_t}$ are $N_t$ target candidates for the drugs treating the disease. Information modeled for target candidates can include interaction, structural, and sequence information.

**Biological context set.** The biological context set includes the cell-type-specific contexts in which the target candidate set operates. This set is denoted as:

$$\mathbb{C} = \{c_1, \ldots, c_{N_c}\}, \tag{16}$$

where $c_1, \ldots, c_{N_c}$ are $N_c$ biological contexts on which drug-target interactions are being evaluated. Information modeled for cell-type-specific biological contexts can include gene expression and tissue hierarchy. The set is constrained to disease-specific cell types and tissues.

**Drug-target identification.** Drug-Target Identification is a binary label $y \in \{1, 0\}$, where $y = 1$ indicates the protein is a candidate therapeutic target. At the same time, 0 means the protein is not such a target.

The goal is to train a model $f_\theta$ for predicting the probability $\hat{y} \in [0, 1]$ that a protein is a candidate therapeutic target in a specific cell type. The model learns an estimator for a disease-specific function of a protein target $t \in \mathbb{T}$ and a cell-type-specific biological context $c \in \mathbb{C}$ as input, and the model is tasked to predict:

$$\hat{y} = f_\theta(t \in \mathbb{T}, c \in \mathbb{C}). \tag{17}$$

### A.2. `tdc_ml.PerturbOutcome`: Perturbation-Response Prediction

**Motivation.** Understanding and predicting transcriptional responses to genetic or chemical perturbations provides insights into cellular adaptation and response mechanisms. Such predictions can advance therapeutic strategies, as they enable

researchers to anticipate how cells will react to targeted interventions, potentially guiding more effective treatments. Models that have shown promise at this task (Roohani et al., 2023; Hetzel et al., 2022) are limited to either genetic or chemical perturbations without being able to generalize to the other. Approaches that can generalize across chemical and genetic perturbations (Piran et al., 2023; Yu & Welch, 2022) may be unable to generalize to unseen perturbations without modification.

### A.2.1. DATASET AND BENCHMARK

We used the scPerturb (Peidli et al., 2024) datasets to benchmark the generalizability of perturbation-response prediction models across seen/unseen perturbations and cell lines. We benchmark models in genetic and chemical perturbations using metrics measuring intra/inter-cell line and seen/unseen perturbation generalizability. We provide results measuring unseen perturbation generalizability for Gene Perturbation Response Prediction using the scPerturb gene datasets (Norman K562, Replogle K562, Replogle RPE1) (Norman et al., 2019; Replogle et al., 2022) with results shown in Table 5. For Chemical Perturbation Prediction, we evaluated chemCPA utilizing cold splits on perturbation type and showed a significant decrease in performance for 3 of 4 perturbations evaluated in table (6). We have also included Biolord (Piran et al., 2023) and scGen (Lotfollahi et al., 2019) for comprehensive benchmarking on the well-explored perturbation response prediction on seen perturbation types problem. These tests were run on sciPlex2 (Srivatsan et al., 2019).

### A.2.2. SCPERTURB DATASET

The scPerturb dataset is a comprehensive collection of single-cell perturbation data harmonized to facilitate the development and benchmarking of computational methods in systems biology. It includes various types of molecular readouts, such as transcriptomics, proteomics, and epigenomics. scPerturb is a harmonized dataset that compiles single-cell perturbation-response data. This dataset is designed to support the development and validation of computational tools by providing a consistent and comprehensive resource. The data includes responses to various genetic and chemical perturbations, crucial for understanding cellular mechanisms and developing therapeutic strategies. Data from different sources are uniformly pre-processed to ensure consistency. Rigorous quality control measures are applied to maintain high data quality. Features across different datasets are standardized for easy comparison and integration.

**Dataset statistics.** 44 publicly available single-cell perturbation-response datasets. Most datasets have, on average, approximately 3000 genes measured per cell. 100,000+ perturbations.

**Dataset split. Cold Split** and **Random Split** defined on cell lines and perturbation types.

**References.** (Peidli et al., 2024)

**Dataset license.** CC BY 4.0

**Code Sample.**

```python
from tdc_ml.multi_pred.perturboutcome import PerturbOutcome
from pandas import DataFrame
test_loader = PerturbOutcome(
    name=scperturb_drug_AissaBenevolenskaya2021)
testdf = test_loader.get_data()
```

### A.2.3. GENETIC PERTURBATION RESPONSE PREDICTION

We use scPerturb gene datasets (Norman K562, Replogle K562, Replogle RPE1) (Norman et al., 2019; Replogle et al., 2022). In the case of single-gene perturbations, we assessed the models based on the perturbation of experimentally perturbed genes not included in the training data. We used data from two genetic perturbation screens, with 1,543 perturbations for RPE-1 (retinal pigment epithelium) cells and 1,092 for K-562 cells, each involving over 170,000 cells. These screens utilized the Perturb-seq assay, which combines pooled screening with single-cell RNA sequencing to analyze the entire transcriptome for each cell. We trained GEARS separately on each dataset. In addition to an existing deep learning-based model (CPA), we also developed a baseline model (no perturbation), assuming that gene expression remains unchanged after perturbation.

We evaluated the models' performance by calculating the mean squared error between the predicted gene expression after perturbation and the actual post-perturbation expression for the held-out set. Based on the highest absolute differential expression upon perturbation, the top 20 most differentially expressed genes were selected.

*Table 5.* **Unseen genetic perturbation response prediction.** We evaluate GEARS across the top 20 differentially expressed genes, based on the highest absolute differential expression upon perturbation, for MSE (MSE@20DEG). Gene expression was measured in log normalized counts. In single-cell analysis, a standard procedure is to normalize the counts within each cell so that they sum to a specific value (usually the median sum across all cells in the dataset) and then to log transform the values using the natural logarithm (Roohani et al., 2023). For both normalization and ranking genes by differential expression, we utilized Scanpy (Wolf et al., 2018). We used the sc.tl.rank_genes_groups() function with default parameters in Scanpy, which employs a t-test to estimate scores. This function provides a z-score for each gene and ranks genes based on the absolute values of the score. Genes showing a significant level of dropout were not included in this metric.

| Dataset | Tissue | Cell Line | Method | MSE@20DEG |
|---|---|---|---|---|
| Norman K562 | K562 | lymphoblast | no-perturb | $0.341_{\pm0.001}$ |
| Norman K562 | K562 | lymphoblast | CPA | $0.230_{\pm0.008}$ |
| Norman K562 | K562 | lymphoblast | GEARS | $0.176_{\pm0.003}$ |
| Replogle 562 | K562 | lymphoblast | no-perturb | $0.126_{\pm0.000}$ |
| Replogle 562 | K562 | lymphoblast | CPA | $0.126_{\pm0.000}$ |
| Replogle 562 | K562 | lymphoblast | GEARS | $0.109_{\pm0.004}$ |
| Replogle RPE1 | RPE-1 | epithelial | no-perturb | $0.164_{\pm0.000}$ |
| Replogle RPE1 | RPE-1 | epithelial | CPA | $0.162_{\pm0.001}$ |
| Replogle RPE1 | RPE-1 | epithelial | GEARS | $0.110_{\pm0.003}$ |

*Listing 1.* The below code illustrates how to retrieve the train, test, and val splits used for the tdc_ml.PerturbOutcome genetic perturbation benchmark

```
from tdc_ml.benchmark_group import geneperturb_group
group = geneperturb_group.GenePerturbGroup()
train_val = group.get_train_valid_split()
test = group.get_test()
# train your model and test on the test set
group.evaluate(preds)
```

### A.2.4. CHEMICAL PERTURBATION RESPONSE PREDICTION

The dataset consists of four drug-based perturbations from sciPlex2 (Srivatsan et al., 2019; Peidli et al., 2024) (BMS, Dex, Nutlin, SAHA). sciPlex2 contains alveolar basal epithelial cells from the A549 (lung adenocarcinoma), K562 (chronic myelogenous leukemia), and MCF7 (mammary adenocarcinoma) tissues. Results are shown in Table 6. Our experiments rely on the coefficient of determination ($R^2$) as the primary performance measure. We calculate this score by comparing actual measurements with counterfactual predictions for all genes. Assessing all genes is essential to evaluating the decoder's overall performance and understanding the background context. Still, it is also beneficial to determine performance based on top differentially expressed genes (Hetzel et al., 2022). The baseline used discards all perturbation information, adequately measuring the improvement resulting by the models' drug encoding (Hetzel et al., 2022). ChemCPA's performance dropped by an average of 15% across the four perturbations. The maximum drop was 34%. Code for intra/inter cell-line benchmarks for chemical (drug) and genetic (CRISPR) perturbations is in Appendix A.2.4 and Appendix A.2.3, respectively. Using this code, users can evaluate models of their choice on the benchmark.

*Table 6.* **Unseen chemical perturbation response prediction.** We have evaluated chemCPA utilizing cold splits on perturbation type and show a significant decrease in performance for 3 of 4 perturbations evaluated. We have also included Biolord (Piran et al., 2023) and scGen (Lotfollahi et al., 2019) for comparison. The dataset used consists of four chemical (drug) perturbations from sciPlex2 (Peidli et al., 2024) (BMS, Dex, Nutlin, SAHA). sciPlex2 contains alveolar basal epithelial cells from the A549 (lung adenocarcinoma), K562 (chronic myelogenous leukemia), and MCF7 (mammary adenocarcinoma) tissues. Our experiments rely on the coefficient of determination ($R^2$) as the primary performance measure.

| Drug | Method | $R^2$ (seen perturbations) | $R^2$ (unseen perturbations) |
|------|--------|----------------------------|------------------------------|
| BMS | Baseline | $0.620_{\pm 0.044}$ | N/A |
| Dex | Baseline | $0.603_{\pm 0.053}$ | N/A |
| Nutlin | Baseline | $0.628_{\pm 0.036}$ | N/A |
| SAHA | Baseline | $0.617_{\pm 0.027}$ | N/A |
| BMS | Biolord | $0.939_{\pm 0.022}$ | N/A |
| Dex | Biolord | $0.942_{\pm 0.028}$ | N/A |
| Nutlin | Biolord | $0.928_{\pm 0.026}$ | N/A |
| SAHA | Biolord | $0.980_{\pm 0.005}$ | N/A |
| BMS | ChemCPA | $0.943_{\pm 0.006}$ | $0.906_{\pm 0.006}$ |
| Dex | ChemCPA | $0.882_{\pm 0.014}$ | $0.540_{\pm 0.013}$ |
| Nutlin | ChemCPA | $0.925_{\pm 0.010}$ | $0.835_{\pm 0.009}$ |
| SAHA | ChemCPA | $0.825_{\pm 0.026}$ | $0.690_{\pm 0.021}$ |
| BMS | scGen | $0.903_{\pm 0.030}$ | N/A |
| Dex | scGen | $0.944_{\pm 0.018}$ | N/A |
| Nutlin | scGen | $0.891_{\pm 0.032}$ | N/A |
| SAHA | scGen | $0.948_{\pm 0.034}$ | N/A |

*Listing 2.* The below code illustrates how to retrieve the train, test, and val splits used for the tdc_ml.PerturbOutcome chemical perturbation benchmark

```
from tdc_ml.benchmark_group import counterfactual_group
group = counterfactual_group.CounterfactualGroup()
train, val = group.get_train_valid_split(remove_unseen=False)
test = group.get_test()
# train your model and test on the test set
group.evaluate(preds)
```

A.2.5. TDC_ML.PERTURBOUTCOME: PERTURBATION-RESPONSE PROBLEM FORMULATION

PyTDC introduces Perturbation-Response prediction task. The predictive, non-generative task is formalized as learning an estimator for a function of the cell-type-specific gene expression response to a chemical or genetic perturbation, taking a perturbation $p \in \mathbb{P}$, a pre-perturbation gene expression profile from the control set $e_0 \in \mathbb{E}_{\not\vdash}$, and the biological context $c \in \mathbb{C}$ under which the gene expression response to the perturbation is being measured:

$$y = f(p, e_0, c). \tag{18}$$

We center our definition on regression for the cell-type-specific gene expression vector in response to a chemical or genetic perturbation.

**Perturbation set.** The perturbation set includes genetic and chemical perturbations. It is denoted by:

$$\mathbb{P} = \{p_1, \ldots, p_{N_p}\}, \tag{19}$$

where $t_p, \ldots, p_{N_p}$ are $N_p$ evaluated perturbations. Information modeled for genetic perturbations can include the type of perturbation (i.e., knockout, knockdown, overexpression) and target gene(s) of the perturbation. Information modeled for chemical perturbations can include chemical structure (i.e., SMILES, InChl) and concentration and duration of treatment.

**Control set.** The control set includes the unperturbed gene expression profiles. This set is denoted as:

$$\mathbb{E}_{\not\vdash} = \{\vec{e}_{0_1}, \ldots, \vec{e}_{N_{e_0}}\}, \tag{20}$$

where $\vec{e}_{0_1}, \ldots, \vec{e}_{N_{e_0}}$ are $N_{e_0}$ unperturbed gene expression profile vectors. Information models for gene expression profiles can include raw or normalized gene expression counts, transcriptomic profiles, and isoform-specific expression levels.

**Biological context set.** The biological context set includes the cell-type-specific contexts under which the perturbed gene expression profile is measured. It is denoted by:

$$\mathbb{C} = \{c_1, \ldots, c_{N_c}\}, \tag{21}$$

where $c_1, \ldots, c_{N_c}$ are the $N_c$ biological contexts under which perturbations are being evaluated. Information modeled for biological contexts can include cell type or tissue type and experimental conditions (Peidli et al., 2024) as well as epigenetic markers (Agrawal et al., 2023; Nair et al., 2020).

**Perturbation-response readouts.** Perturbation-Response is a gene expression vector $\vec{e_1}$, where $\vec{e_1}_i$ denotes the expression of the i-th gene in the vector. It is the outcome of applying a perturbation, $p_i \in \mathbb{P}$, within a biological context, $c_j \in \mathbb{C}$, to a cell with a measured control gene expression vector, $e_{0_k} \in \mathbb{E}_{\not{p}}$.

The Perturbation-Response Prediction learning task is to learn a regression model $f_\theta$ estimating the perturbation-response gene expression vector $\hat{\vec{e_1}}$ for a perturbation applied in a cell-type-specific biological context to a control:

$$\hat{\vec{e_1}} = f_\theta(p \in \mathbb{P}, e_0 \in \mathbb{E}_{\not{p}}, c \in \mathbb{C}). \tag{22}$$

### A.3. `tdc_ml.ProteinPeptide`: Contextualized Protein-Peptide Interaction Prediction

**Motivation.** Evaluating protein-peptide binding prediction models requires standardized benchmarks, presenting challenges in assessing and validating model performance across different studies (Romero-Molina et al., 2022). Despite the availability of several benchmarks for protein-protein interactions, this is not the case for protein-peptide interactions. The renowned multi-task benchmark for Protein sEquence undERstanding (PEER) (Xu et al., 2022) and MoleculeNet (Wu et al., 2018) both lack support for a protein-peptide interaction prediction task. Furthermore, protein-peptide binding mechanisms vary wildly by cellular and biological context (Janeway et al., 2001; Murphy & Weaver, 2016; Ploegh, 1998; Schatz & Swanson, 2011). Current models, as such, tend to be restricted to one task instance (i.e., T Cell Receptor (TCR) and Peptide-MHC Complex or B Cell Receptor (BCR) and Antigen Peptide binding) and do not span protein-peptide interactions (Springer et al., 2019; Chen et al., 2020b; Liu et al., 2021; Bradley, 2022; Hu & Liu, 2019; Lupia et al., 2019; Saxena et al., 2020).

We define and evaluate a subtask for TCR-Epitope binding interaction prediction applying contextual AI (Section A.3.5) to the T Cell cell line.

#### A.3.1. TCR-EPITOPE (PEPTIDE-MHC COMPLEX) INTERACTION PREDICTION

The critical challenge in TCR-Epitope (Peptide-MHC Complex) interaction prediction lies in creating a model that can effectively generalize to unseen TCRs and epitopes (Moris et al., 2020). While TCR-H (T et al., 2023) and TEINet (Jiang et al., 2023) have shown improved performance on prediction for known epitopes, by incorporating advanced features like attention mechanisms and transfer learning, the performance considerably drops for unseen epitopes (Cai et al., 2022; Weber et al., 2021). Another challenge in TCR-Epitope interaction prediction lies in the choice of heuristic for generating negative samples, with non-binders often underrepresented or biased in curated datasets, leading to inaccurate predictions when generalized (Kim et al., 2023).

**Datasets and Benchmarks.** PyTDC establishes a curated dataset and benchmark within its single-cell protein-peptide binding affinity prediction task to measure model generalizability to unseen TCRs and epitopes and model sensitivity to the selection of negative data points. Benchmarking datasets use three types of heuristics for generating negative samples: random shuffling of epitope and TCR sequences (RN), experimental negatives (NA), and pairing external TCR sequences with epitope sequences (ET). We harness data from the TC-hard dataset (Grazioli et al., 2022b) for the first two types and PanPep (Gao et al., 2023) for the third type. Both datasets use hard (Grazioli et al., 2022b) splits, ensuring that epitopes in the testing set are not present in the training set. Our results (Table 7) show the lack of a reasonable heuristic for generating negative samples, with model performance evaluation shown to be unsatisfactory. For two heuristics, all models perform poorly. The best-performing model in ET is MIX-TPI, with roughly 0.70 AUROC. The best-performing model in RN is AVIB-TCR, with approximately 0.576 AUROC. For NA, 4 of 6 models perform near-perfectly as measured on AUROC. Models benchmarked include AVIB-TCR (Grazioli et al., 2022a), MIX-TPI (Yang et al., 2023), Net-TCR2 (Jensen & Nielsen, 2023), PanPep (Moris et al., 2020), TEINet (Jiang et al., 2023), and TITAN (Weber et al., 2021).

*Table 7.* **TCR-epitope binding interaction binary classification performance.** All models perform poorly under realistic but challenging RN and ET experimental setups. The best-performing model in RN is AVIB-TCR, with an average of 0.576 (AUROC). The best-performing model in ET is MIX-TPI, with an average of 0.700 (AUROC). For NA, 4 of 6 models achieve near-perfect AUROC.

| Methods | Experimental setup | ACC | F1 | AUROC | AUPRC |
|---------|--------------------|------|------|-------|-------|
| AVIB-TCR | RN | $0.570\pm0.028$ | $0.468\pm0.086$ | $0.576\pm0.049$ | $0.605\pm0.044$ |
| MIX-TPI | RN | $0.539\pm0.039$ | $0.408\pm0.122$ | $0.558\pm0.028$ | $0.597\pm0.049$ |
| Net-TCR2 | RN | $0.528\pm0.050$ | $0.354\pm0.036$ | $0.551\pm0.042$ | $0.554\pm0.075$ |
| PanPep | RN | $0.507\pm0.028$ | $0.473\pm0.039$ | $0.535\pm0.021$ | $0.579\pm0.040$ |
| TEINet | RN | $0.459\pm0.036$ | $0.619\pm0.036$ | $0.535\pm0.029$ | $0.581\pm0.043$ |
| TITAN | RN | $0.476\pm0.063$ | $0.338\pm0.111$ | $0.502\pm0.066$ | $0.523\pm0.055$ |
| AVIB-TCR | ET | $0.611\pm0.012$ | $0.553\pm0.020$ | $0.683\pm0.010$ | $0.815\pm0.006$ |
| MIX-TPI | ET | $0.652\pm0.009$ | $0.523\pm0.035$ | $0.703\pm0.016$ | $0.825\pm0.014$ |
| Net-TCR2 | ET | $0.621\pm0.027$ | $0.522\pm0.020$ | $0.674\pm0.017$ | $0.810\pm0.016$ |
| PanPep | ET | $0.556\pm0.009$ | $0.506\pm0.011$ | $0.638\pm0.009$ | $0.753\pm0.009$ |
| TEINet | ET | $0.356\pm0.008$ | $0.512\pm0.010$ | $0.571\pm0.009$ | $0.646\pm0.011$ |
| TITAN | ET | $0.670\pm0.013$ | $0.492\pm0.048$ | $0.624\pm0.021$ | $0.733\pm0.018$ |
| AVIB-TCR | NA | $0.636\pm0.062$ | $0.197\pm0.169$ | $0.944\pm0.021$ | $0.949\pm0.023$ |
| MIX-TPI | NA | $0.952\pm0.029$ | $0.937\pm0.040$ | $0.992\pm0.002$ | $0.995\pm0.001$ |
| Net-TCR2 | NA | $0.655\pm0.051$ | $0.274\pm0.123$ | $0.973\pm0.009$ | $0.985\pm0.005$ |
| PanPep | NA | $0.419\pm0.011$ | $0.352\pm0.006$ | $0.611\pm0.014$ | $0.499\pm0.031$ |
| TEINet | NA | $0.413\pm0.023$ | $0.582\pm0.023$ | $0.973\pm0.011$ | $0.981\pm0.006$ |
| TITAN | NA | $0.695\pm0.050$ | $0.404\pm0.141$ | $0.629\pm0.053$ | $0.661\pm0.040$ |

### A.3.2. TCHARD DATASET

The TChard dataset is designed for TCR-peptide/-pMHC binding prediction. It includes over 500,000 samples from sources such as IEDB, VDJdb, McPAS-TCR, and the NetTCR-2.0 repository. The dataset is utilized to investigate how state-of-the-art deep learning models generalize to unseen peptides, ensuring that test samples include peptides not found in the training set. This approach highlights the challenges deep learning methods face in robustly predicting TCR recognition of peptides not previously encountered in training data.

**Dataset statistics.** 500,000 samples

**Dataset split.** Cold Split referred to as "Hard" split in (Grazioli et al., 2022b).

**References.** (Grazioli et al., 2022b)

**Dataset license.** Non-Commercial Use

**Code Sample.**

```
from tdc_ml.resource.dataloader import DataLoader
data = DataLoader(name=tchard)
self.split = data.get_split()
```

### A.3.3. PANPEP DATASET

PanPep is a framework constructed in three levels for predicting the peptide and TCR binding recognition. We have provided the trained meta learner and external memory, and users can choose different settings based on their data available scenarios: Few known TCRs for a peptide: few-shot setting; No known TCRs for a peptide: zero-shot setting; plenty of known TCRs for a peptide: majority setting. More information is available in the Github repo .

**Dataset statistics.** Data from multiple studies involving millions of TCR sequences.

**Dataset split. Cold Split** referred to as "Hard" split in (Gao et al., 2023).

**References.** (Gao et al., 2023)

**Dataset license.** GPL-3.0

**Code Sample.**

```
from tdc_ml.resource.dataloader import DataLoader
```

```
data = DataLoader(name=panpep)
self.split = data.get_split()
```

### A.3.4. (YE X ET AL) DATASET

Affinity selection-mass spectrometry data of discovered ligands against single biomolecular targets (MDM2, ACE2, 12ca5) from the Pentelute Lab of MIT This dataset contains affinity selection-mass spectrometry data of discovered ligands against single biomolecular targets. Several AS-MS-discovered ligands were taken forward for experimental validation to determine the binding affinity (KD) as measured by biolayer interferometry (BLI) to the listed target protein. If listed as a "putative binder," AS-MS alone was used to isolate the ligands to the target, with KD ¡ 1 uM required and often observed in orthogonal assays, though there is some (¡ 50%) chance that the ligand is nonspecific. Most of the ligands are putative binders, with 4446 total provided. For those characterized by BLI (only 34 total), the average KD is 266 ± 44 nM; the median KD is 9.4 nM.

**Dataset statistics.** 34 positive ligands, 4446 putative binders, and three proteins

**Dataset Split. Stratified Split** and **N/A Split**: We provide stratified 10/90 split on train/test as well as "test set only" split.

**References.** (Ye et al., 2022; Brown et al., 2023)

**Dataset license.** CC BY 4.0

**Code Sample.**

```
from tdc_ml.multi_pred import ProteinPeptide
data = ProteinPeptide(name=brown_mdm2_ace2_12ca5)
data.get_split()
```

*Listing 3.* The below code illustrates how to retrieve the train, test, and val splits used for the tdc_ml.TCREpitope benchmark

```
from tdc_ml.benchmark_group.tcrepitope_group import TCREpitopeGroup
group = TCREpitopeGroup()
train_val = group.get_train_valid_split()
test = group.get_test()
# train your model and test on the test set
group.evaluate(preds)
```

### A.3.5. TDC_ML.PROTEINPEPTIDE: PROTEIN-PEPTIDE INTERACTION PREDICTION PROBLEM FORMULATION

PyTDC introduces the Protein-Peptide Binding Affinity prediction task. The predictive, non-generative task is to learn a model estimating a function of a protein, peptide, antigen processing pathway, biological context, and interaction features. It outputs a binding affinity value (e.g., dissociation constant Kd, Gibbs free energy $\Delta G$) or binary label indicating strong or weak binding. The binary label can also include additional biomarkers, such as allowing for a positive label if and only if the binding interaction is specific (Brown et al., 2023; Hoofnagle & Resing, 2001; Smith & Kelleher, 2013). To account for additional biomarkers beyond binding affinity value, our task is specified with a binary label.

**Protein set.** The protein set includes target proteins. It is denoted by:

$$\mathbb{P} = \{p_1, \ldots, p_{N_p}\}, \tag{23}$$

where $p_1, \ldots, p_{N_p}$ are $N_p$ target proteins. Information modeled for proteins can include sequence, structural, or post-translational modification data.

**Peptide set.** The control set includes the peptide candidates. This set is denoted as:

$$\mathbb{S} = \{s_1, \ldots, s_{N_s}\}, \tag{24}$$

where $s_1, \ldots, s_{N_s}$ are $N_s$ candidate peptides. Information modeled for candidate peptides can include sequence, structural, and physicochemical data.

**Antigen processing pathway set.** The antigen processing pathway set includes antigen processing pathway profile information about prior steps in the biological antigen presentation pathway processes. It is denoted by:

$$\mathbb{A} = \{a_1, \ldots, a_{N_a}\},$$ (25)

where $a_1, \ldots, a_{N_a}$ are the $N_a$ antigen processing pathway profiles modeled. Information modeled in a profile can include proteasomal cleavage sites (Nielsen et al., 2005), classification into viral, bacterial, and self-protein sources and endogenous vs exogenous processing pathway (Reynisson et al., 2020; O'Donnell et al., 2020; Saxena et al., 2020; Boulanger et al., 2018), and target/receptor-specific pathway attributes such as transporter associated with antigen processing (TAP) affinity (Bhasin et al., 2007), and endosomal/lysosomal processing efficiency (Koşaloğlu-Yalçın et al., 2020).

**Interaction set.** It contains the interaction feature profiles. The set is denoted by:

$$\mathbb{I} = \{i_1, \ldots, i_{N_i}\},$$ (26)

where $i_1, \ldots, i_{N_i}$ are the $N_i$ interaction feature profiles. Information modeled in an interaction feature profile can include contact maps (Huang & Ding, 2022; Yaseen et al., 2018; Li et al., 2021; You & Shen, 2020), distance maps (Yaseen et al., 2018; Zheng et al., 2019), electrostatic interactions (Huang & Ding, 2022), and hydrogen bonds (Huang & Ding, 2022).

**Cell-type-specific biological context set.** It contains the interaction feature profiles. The set is denoted by:

$$\mathbb{C} = \{c_1, \ldots, c_{N_c}\},$$ (27)

where $c_1, \ldots, c_{N_c}$ are the $N_c$ cell-type-specific biological contexts under which the protein-peptide interaction is being evaluated. Information modeled in the cell-type-specific biological context can include transcriptomic and proteomic data. We note, however, that, to our knowledge, single-cell transcriptomic and proteomic data has yet to be used in protein-peptide binding affinity prediction, outlining a promising avenue of research in developing machine learning models for peptide-based therapeutics.

**Protein-peptide interaction.** It is a binary label, $y \in \{1, 0\}$, where $y = 1$ indicates a protein-peptide pair met the target biomarkers and $y = 0$ indicates the pair did not meet the target biomarkers.

The Protein-Peptide Interaction Prediction learning task is to learn a binary classification model $f_\theta$ estimating the probability, $\hat{y}$, of a protein-peptide interaction meeting specific biomarkers:

$$\hat{y} = f_\theta(p \in \mathbb{P}, s \in \mathbb{S}, a \in \mathbb{A}, i \in \mathbb{I}, c \in \mathbb{C}).$$ (28)

### A.4. tdc_ml.TrialOutcome

PyTDC introduces a model framework, task definition, dataset, and benchmark for the Clinical Outcome Prediction task tailored to precision medicine. The framework and definition aim to assess clinical trials systematically and comprehensively by predicting various endpoints for patient sub-populations. Our benchmark uses the Trial Outcome Prediction (TOP) dataset (Fu et al., 2022). TOP consists of 17,538 clinical trials with 13,880 small-molecule drugs and 5,335 diseases.

**Dataset and benchmark.** Our benchmark uses the Trial Outcome Prediction (TOP) dataset (Fu et al., 2022). TOP consists of 17,538 clinical trials with 13,880 small-molecule drugs and 5,335 diseases. Out of these trials, 9,999 (57.0%) succeeded (i.e., meeting primary endpoints), and 7,539 (43.0%) failed. Out of these trials, 1,787 were in Phase I testing (toxicity and side effects), 6,102 in Phase II (efficacy), and 4,576 in Phase III (effectiveness compared to current standards). We perform a temporal split for benchmarking. The train/validation and test are time-split by the date January 1, 2014, i.e., the start dates of the test set are after January 1, 2014, while the completion dates of the train/validation set are before January 1, 2014. Here, the HINT model (Fu et al., 2022), is benchmarked against COMPOSE (Gao et al., 2020) and DeepEnroll (Zhang et al., 2020) models. Results are shown in table 8.

*Table 8.* Clinical Trial Outcome Prediction task benchmark model results on the TOP dataset (Fu et al., 2022), described in section A.4 (Velez-Arce et al., 2024).

| Model | Phase 1 AUPRC | Phase 2 AUPRC | Phase 3 AUPRC | Indication Level AUPRC |
|---|---|---|---|---|
| HINT | 0.772 | 0.607 | 0.623 | 0.703 |
| COMPOSE | 0.665 | 0.532 | 0.545 | 0.624 |
| DeepEnroll | 0.701 | 0.580 | 0.590 | 0.655 |

A.4.1. TOP DATASET

TOP (Fu et al., 2022) consists of 17,538 clinical trials with 13,880 small-molecule drugs and 5,335 diseases. Out of these trials, 9,999 (57.0%) succeeded (i.e., meeting primary endpoints), and 7,539 (43.0%) failed. For each clinical trial, we produce the following four data items: (1) drug molecule information, including Simplified Molecular Input Line Entry System (SMILES) strings and molecular graphs for the drug candidates used in the trials; (2) disease information including ICD-10 codes (disease code), disease description, and disease hierarchy in terms of CCS codes (https://www.hcup-us.ahrq.gov/toolssoftware/ccs10/ccs10.jsp); (3) trial eligibility criteria are in unstructured natural language and contain inclusion and exclusion criteria; and (4) trial outcome information includes a binary indicator of trial success (1) or failure (0), trial phase, start and end date, sponsor, and trial size (i.e., number of participants).

**Dataset statistics.** Phase I: 2,402 trials / Phase II: 7,790 trials / Phase III: 5,741 trials.

**Dataset split. Temporal Split** as defined in (Fu et al., 2022) and Section A.4.

**References.** (Fu et al., 2022)

**Dataset license.** Non-Commercial Use

**Code Sample.**

```
from tdc_ml.multi_pred import TrialOutcome
data = TrialOutcome(name = 'phase1') # 'phase2' / 'phase3'
split = data.get_split()
```

A.4.2. CLINICAL TRIAL OUTCOME PREDICTION PROBLEM FORMULATION

The Clinical Trial Outcome Prediction task is formulated as a binary classification problem, where the machine learning model predicts whether a clinical trial will have a positive or negative outcome. It is a function that takes patient data, trial design, treatment characteristics, disease, and macro variables as inputs and outputs a trial outcome prediction, a binary indicator of trial success (1) or failure (0).

**Patient set.** The patient set includes one or multiple patient sub-populations, with the extreme case representing personalization. It is denoted as follows:

$$\mathbb{P} = \{p_1, \ldots, p_{N_p}\}, \tag{29}$$

where $p_1, \ldots, p_{N_p}$ are $N_p$ patient sub-populations in this trial. The TOP benchmark (Fu et al., 2022) dataset represents patient data as part of the trial eligibility criteria. Patient data can include demographics (Hong et al., 2021; Lv et al., 2020; Rongali et al., 2020; Rahimian et al., 2018; Park et al., 2020), baseline health metrics (Rahimian et al., 2018; Park et al., 2020; Bedon et al., 2021), and medical history (Hong et al., 2021; Lv et al., 2020; Rongali et al., 2020; Rahimian et al., 2018; Park et al., 2020).

**Trial design set.** The trial design set includes this clinical trial's design profiles. It is denoted as:

$$\mathbb{D} = \{d_1, \ldots, d_{N_d}\}, \tag{30}$$

where $d_1, \ldots, d_{N_d}$ are $N_d$ eligible trial design profiles for this clinical trial. Trial design profiles can model information including phase of the trial (Fu et al., 2022), number of participants, duration of the trial, trial eligibility criteria (Fu et al., 2022), and randomization and blinding methods (Schperberg et al., 2020; Wang et al., 2021b; Dai et al., 2022).

**Treatment set.** The treatment set includes the candidate treatments for the trial. It is denoted as:

$$\mathbb{T} = \{t_1, \ldots, t_{N_t}\}, \tag{31}$$

where $t_1, \ldots, t_{N_t}$ are $N_t$ candidate treatments for the clinical trial. The information modeled for treatments can include type of treatment (drug (Fu et al., 2022; Brbi et al., 2024), device (Kalscheur et al., 2018; Fujima et al., 2019; van Os et al., 2018), procedure (Asadi et al., 2014; Arvind et al., 2018; Senders et al., 2018; Jourahmad et al., 2023; MacKay et al., 2021)), dosage and administration route (Schperberg et al., 2020; Bedon et al., 2021; Bowman et al., 2023), mechanism of action (Sayaman et al., 2020; Beacher et al., 2021; Siah et al., 2019), pre-clinical and early-phase trial results (Beacher et al., 2021; Bedon et al., 2021; Beinse et al., 2019; Wang et al., 2023).

**Macro context set.** The macro context set contains the configurations of macro variables relevant to the clinical trial. It is denoted as:

$$\mathbb{C} = \{c_1, \ldots, c_{N_c}\}, \tag{32}$$

where $c_1, \ldots, c_{N_c}$ are $N_c$ configurations containing the values for macro variables relevant to the trial, which can include geography (Shamout et al., 2020; Beacher et al., 2021; Wang et al., 2023; Liu & Salinas, 2017) and regulatory considerations (Beacher et al., 2021; Shamout et al., 2020).

**Trial outcomes.** The trial outcome is a binary label $y \in \{1, 0\}$, where $y = 1$ indicates the trial met their primary endpoints, while 0 means failing to meet with the primary endpoints.

The learning task is to learn a model $f_\theta$ for predicting the trial success probability $\hat{y}$, where $\hat{y} \in [0, 1]$:

$$\hat{y} = f_\theta(p \in \mathbb{P}, d \in \mathbb{D}, t \in \mathbb{T}, c \in \mathbb{C}). \tag{33}$$

### A.5. tdc_ml.sBDD Structure-Based Drug Design

Structure-based drug design aims to create diverse new molecules that bind to protein pockets (3D structures) and have favorable chemical properties. These attributes are evaluated using pharmaceutically-relevant oracle functions. In this task, an ML model learns molecular traits of protein pockets from a comprehensive dataset of protein-ligand pairs. Subsequently, potential new molecules can be generated using the acquired conditional distribution. The generated molecules must exhibit outstanding properties, including high binding effectiveness and structural variety. They must meet other user-specified criteria, such as the feasibility of synthesis (synthesizability/designability) and similarity to known drugs.

#### A.5.1. PDBBIND DATASET

PDBBind is a comprehensive database extracted from PDB with experimentally measured binding affinity data for protein-ligand complexes. PDBBind does not allow the dataset to be re-distributed in any format. Thus, we could not host it on the TDC server. However, we provide an alternative route since significant processing is required to prepare the dataset ML. The user only needs to register at http://www.pdbbind.org.cn/, download the raw dataset, and then provide the local path. TDC will then automatically detect the path and transform it into an ML-ready format for the TDC data loader.

**Dataset statistics.** 19,445 protein-ligand pairs

**Dataset split. Random Split**

**References.** (Liu et al., 2015)

**Dataset license.** See note in the description on the TDC website.

**Code Sample.**

```
from tdc_ml.generation import SBDD
data = SBDD(name='PDBBind', path='./pdbbind')
split = data.get_split()
```

#### A.5.2. DUD-E DATASET

DUD-E provides a directory of valuable decoys for protein-ligand docking.

**Dataset statistics.** 22,886 active compounds and affinities against 102 targets. DUD-E does not support pocket extraction as protein and ligand are not aligned.

**Dataset split. Random Split**

**References.** (Mysinger et al., 2012)

**Dataset license.** Not specified

**Code Sample.**

```
from tdc_ml.generation import SBDD
data = SBDD(name='dude')
split = data.get_split()
```

#### A.5.3. SCPDB DATASET

scPDB is processed from PDB for structure-based drug design that identifies suitable binding sites for protein-ligand docking.

**Dataset statistics.** 16,034 protein-ligand pairs over 4,782 proteins and 6,326 ligands

**Dataset split. Random Split**

**References.** (Meslamani et al., 2011)

**Dataset license.** Not specified

**Code Sample.**

```
from tdc_ml.generation import SBDD
data = SBDD(name='scPDB')
split = data.get_split()
```

A.5.4. STRUCTURE-BASED DRUG DESIGN PROBLEM FORMULATION

Structure-based Drug Design aims to generate diverse, novel molecules with high binding affinity to protein pockets (3D structures) and desirable chemical properties. Oracle functions measure these properties. A machine learning task first learns the molecular characteristics given specific protein pockets from a large set of protein-ligand pair data. Then, from the learned conditional distribution, we can sample novel candidates.

**Target candidate set.** The target candidate set includes proteins, nucleic acids, or other biomolecules drugs can interact with, producing a therapeutic effect or causing a biological response. It is denoted by:

$$\mathbb{T} = \{t_1, \ldots, t_{N_t}\}, \tag{34}$$

where $t_1, \ldots, t_{N_t}$ are $N_t$ target candidates for the evaluated set of drugs. Information modeled for target candidates can include interaction, structural, and sequence information.

**Ligand candidate set.** The ligand drug candidate set includes the drug molecules being tested for a particular therapeutic effect or biological response. It is denoted by:

$$\mathbb{L} = \{l_1, \ldots, l_{N_l}\}, \tag{35}$$

where $l_1, \ldots, l_{N_l}$ are the $N_l$ ligand/drug molecules being evaluated. Drug modeling can include molecular structure, often represented in formats such as SMILES (Simplified Molecular Input Line Entry System) or InChI (International Chemical Identifier) (Öztürk et al., 2019), physicochemical properties like hydrophobicity and molecular weight (Chen et al., 2020b), and molecular descriptors and fingerprints (Wang et al., 2019).

**Scoring function.** The scoring function, denoted by $S$, evaluates the binding affinity of ligand $l \in \mathbb{L}$ to protein target $t \in \mathbb{T}$.

**Drug-likeness function.** Function representing the drug-likeness of ligand $l \in \mathbb{L}$, including properties like solubility, stability, and toxicity.

The generative learning task is to generate the ligand $l \in \mathbb{L}$ maximizing binding affinity, $S_\theta$, and drug-likeness, $f_\theta$. Given a loss function, $\text{Loss}(S(t, l), f(l))$, for $t \in \mathbb{T}$ and $l \in \mathbb{L}$, the first step is to learn a model $M_\theta$ s.t.,

$$M_\theta = \text{argmin}_\theta[\text{Loss}(S_\theta(t, l), f_\theta(l))]. \tag{36}$$

This is followed by the ligand optimization step, which optimizes the ligand for maximum binding affinity and drug-likeness given the trained model. A ligand optimization function, $F$, such as addition or multiplication, is used for the optimization:

$$l^* = \text{argmax}_{l \in \mathbb{L}}[F(S_\theta(t \in \mathbb{T}, l), f_\theta(l))]. \tag{37}$$

An example formulation would be as follows:

$$l^* = \text{argmax}_{l \in \mathbb{L}}[S_\theta(t \in \mathbb{T}, l) \times f_\theta(l)]. \tag{38}$$

# B. PyTDC Architecture

## B.1. API-First Design and Model-View-Controller

See section 3.1 in the main text. PyTDC (Velez-Arce et al., 2024) drastically expands dataset retrieval capabilities available in TDC-1 beyond those of other leading benchmarks. Leading benchmarks, like MoleculeNet (Wu et al., 2018) and

TorchDrug (Zhu et al., 2022) have traditionally provided dataloaders to access file dumps. PyTDC introduces API-integrated multimodal data-views (Churi et al., 2016; Emmerik et al., 1993; Rammerstorfer & Mössenböck, 2003). To do so, the software architecture of PyTDC was redesigned using the Model-View-Controller (MVC) design pattern (Bucanek, 2009; Malik et al., 2021). The MVC architecture separates the model (data logic), view (UI logic), and controller (input logic), which allows for the integration of heterogeneous data sources and ensures consistency in data views (Churi et al., 2016). The MVC pattern supports the integration of multiple data modalities by using data mappings and views (Rammerstorfer & Mössenböck, 2003). The MVC-enabled-multimodal retrieval API is powered by PyTDC's Resource Model (Section B.2).

### B.1.1. TDC DATALOADER (*Model*)

As per the TDC-1 specification, this component queries the underlying data source to provide raw or processed data to upstream function calls. We augmented this component beyond TDC-1 functionality to allow for querying datasets introduced in PyTDC, such as the CZ CellXGene.

### B.1.2. TDC MEANINGFUL DATA SPLITS AND MULTIMODAL DATA PROCESSING (*View*)

As per the TDC-1 specification, this component implements data splits to evaluate model generalizability to out-of-distribution samples and data processing functions for multiple modalities. We augmented this component to act on data views (Churi et al., 2016) specified by PyTDC's controller.

### B.1.3. PyTDC DOMAIN-SPECIFIC LANGUAGE (*Controller*)

PyTDC develops an Application-Embedded Domain-Specific Data Definition Programming Language facilitating the integration of multiple modalities by generating data views from a mapping of multiple datasets and functions for transformations, integration, and multimodal enhancements, while mantaining a high level of abstraction (Membarth et al., 2016) for the Resource framework. We include examples developing multimodal datasets leveraging this MVC DSL in listing 4.

### B.2. Resource Model

The Commons introduces a redesign of TDC-1's dataset layer into a new data model dubbed the PyTDC resource, which has been developed under the MVC paradigm to integrate multiple modalities into the API-first model of PyTDC.

### B.2.1. CZ CELLXGENE WITH SINGLE CELL BIOLOGY DATASETS

CZ CellXGene (CZI Single-Cell Biology, et al., 2023) is a open-source platform for analysis of single-cell RNA sequencing data. We leverage the CZ CellXGene to develop a PyTDC Resource Model for constructing large-scale single-cell datasets that maps gene expression profiles of individual cells across tissues, healthy and disease states. PyTDC leverages the SOMA (Stack of Matrices, Annotated) API, adopts TileDB-SOMA (Papadopoulos et al., 2016) for modeling sets of 2D annotated matrices with measurements of features across observations, and enables memory-efficient querying of single-cell modalities (i.e., scRNA-seq, snRNA-seq), across healthy and diseased samples, with tabular annotations of cells, samples, and patients the samples come from.

We develop a remote procedure call (RPC) API taking the string name (e.g., listing 5 in section B.3.1) of the desired reference dataset as specified in the CellXGene (CZI Single-Cell Biology, et al., 2023). The remote procedure call for fetching data is specified as a Python generator expression, allowing the user to iterate over the constructed single-cell atlas without loading it into memory (Tasoluk & Tanrikulu, 2023). Specifying the RPC as a Python generator expression allows us to make use of memory-efficient querying as provided by TileDB (Papadopoulos et al., 2016). The single cell datasets can be integrated with therapeutics ML workflows in PyTDC by using tools such as PyTorch's IterableDataset module (Paszke et al., 2019).

**Knowledge graph, external APIs, and model hub.** We have developed a framework for biomedical knowledge graphs to enhance multimodality of dataset retrieval via PyTDC's Resource Model. Our system leverages PrimeKG to integrate 20 high-quality resources to describe 17,080 diseases with 4,050,249 relationships (Chandak et al., 2023). Our framework also extends to external APIs, with data views currently leveraging BioPython (Cock et al., 2009), for obtaining nucleotide sequence information for a given non-coding RNA ID from NCBI (Cock et al., 2009), and The Uniprot Consortium's RESTful GET API (Consortium, 2021) for obtaining amino acid sequences. In addition we've developed the framework to allow access to embedding models under diverse biological contexts via the PyTDC Model Hub. Examples using these

components are in sections B.3.2 and B.3.3.

## B.3. API-first design code samples

We provide code samples for the components described in sections 3.1, B.1, and B.2.

*Listing 4.* The above configuration augments a protein-peptide dataset with an additional modality, amino acid sequence, and invokes numerous data processing functions tailored to the specific needs of the underlying dataset. Added information for this demonstration can be found at: . There are more complex workflows implemented for current TDC dataviews and all such views leveraging the DSL can be found in the repo at

```
from .config import DatasetConfig
from ..feature_generators.protein_feature_generator import
    ↪ ProteinFeatureGenerator

class BrownProteinPeptideConfig(DatasetConfig):
    Configuration for the brown-protein-peptide datasets

    def __init__(self):
        super(BrownProteinPeptideConfig, self).__init__(
            dataset_name=brown_mdm2_ace2_12ca5,
            data_processing_class=ProteinFeatureGenerator,
            functions_to_run=[
                autofill_identifier, create_range, insert_protein_sequence
            ],
            args_for_functions=[{
                autofill_column: Name,
                key_column: Sequence,
            }, {
                column: KD (nM),
                keys: [Putative binder],
                subs: [0]
            }, {
                gene_column: Protein Target
            }],
            var_map={
                X1: Sequence,
                X2: protein_or_rna_sequence,
                ID1: Name,
                ID2: Protein Target,
            },
        )
```

### B.3.1. PYTDC MULTIMODAL SINGLE-CELL RETRIEVAL API

We focus on the use case of an ML researcher who wishes to train a model on a large-scale single-cell atlas. In particular, researchers would be familiar with and have trained models on traditional single-cell datasets such as Tabula Sapiens (Jones et al., 2022). Their interest is to scale a model by training it on a more extensive single-cell atlas based on this reference dataset. We build such an API. Specifically, given a reference dataset available in CellXGene Discover (CZI Single-Cell Biology, et al., 2023), we allow the user to perform a memory-efficient query using TileDB-SOMA to expand the reference dataset to include cell entries with non-zero readouts for any of the genes present in the reference dataset. This allows users to build large-scale single-cell atlases on familiar reference datasets. The example below illustrates how a user may construct a large-scale atlas with Tabula Sapiens as the reference dataset. Other use cases include augmenting datasets using knowledge graphs and cell-type-specific biomedical contexts. These capabilities are all powered by the

Model-View-Controller framework (section B.1).

*Listing 5.* The example below illustrates how a user may construct a large-scale atlas with Tabula Sapiens as the reference dataset using the PyTDC CELLXGENE API.

```python
from tdc_ml.multi_pred.single_cell import CellXGene
dataloader = CellXGene(name=Tabula_Sapiens_-_All_Cells)
gen = dataloader.get_data(
    value_filter=tissue_==_'brain'_and_sex_==_'male'
)
df = next(gen)
```

*Listing 6.* In addition to our PyTDC DataLoader API implementation for the CellXGene RPC API, we provide a simplified wrapper over the CellXGene Census Discovery API, which allows users to perform remote procedure calls to fetch Cell Census data in more machine-learning-friendly formats like Pandas and Scipy. We also maintain support for the AnnData format. Users can query Cell Census counts as well as metadata using this API. The code sample below illustrates such usage.

```python
from tdc_ml.resource import cellxgene_census

# initialize Census Resource and query filters
resource = cellxgene_census.CensusResource()
cell_value_filter = tissue_==_'brain'_and_sex_==_'male'
cell_column_names = [assay, cell_type, tissue]

# Obtaining cell metadata from the cellxgene census in pandas format
obsdf = resource.get_cell_metadata(
    value_filter=cell_value_filter,
    column_names=cell_column_names,
    fmt=pandas)
```

### B.3.2. PRIMEKG KNOWLEDGE GRAPH

PrimeKG supports drug-disease prediction by including an abundance of 'indications,' 'contradictions', and 'off-label use' edges, which are usually missing in other knowledge graphs. We accompany PrimeKG's graph structure with text descriptions of clinical guidelines for drugs and diseases to enable multimodal analyses (Chandak et al., 2023). The code below depict example use cases of the PyTDC PrimeKG API. Demonstrations are additionally available in https://colab.research.google.com/drive/1kYH8nt3nW7tXYBPNcfYuDbWxGTqOEnWg?usp=sharing.

*Listing 7.* We illustrate here example utilities for retrieving drug-target-disease associations using the PyTDC PrimeKG API

```python
from tdc_ml.resource import PrimeKG

pkg = PrimeKG()
pkgdf = pkg.get_data()

def get_all_drug_evidence(disease):
    given_a_disease,_retrieve_all_drugs_interacting_with_proteins_relevant_to_
        ↪ disease
    prots = pkgdf[(pkgdf[relation] == disease_protein) & (pkgdf[x_name] == disease
        ↪ )][y_name].unique()
    drugs = pkgdf[(pkgdf[relation] == drug_protein) & (pkgdf[y_name].isin(prots))]
    relations = drugs[display_relation].unique()
    out = {}
    for rel in relations:
        out[rel] = drugs[drugs[display_relation] == rel][x_name].unique()
```

```python
    return out

def get_all_associated_targets(disease):
    return pkgdf[(pkgdf[relation] == disease_protein) & (pkgdf[x_name] == disease)
        ↪ ][[y_name, display_relation]]

def get_disease_disease_associations(disease):
    return pkgdf[(pkgdf[relation] == disease_disease) & (pkgdf[x_name] == disease)
        ↪ ][[y_name, display_relation]]

def get_labels_from_evidence(disease):
    diseases = get_disease_disease_associations(disease)[y_name]
    out = set()
    for d in diseases:
        targets = get_all_associated_targets(d)[y_name].unique()
        out.update(targets)
    return list(out)

def all_diseases_by_keyword(kw):
    return pkgdf[(pkgdf[relation] == disease_protein) & (pkgdf[x_name].str.
        ↪ contains(kw, case=False, na=False))][x_name].unique()

if __name__ == __main__:
    x = all_diseases_by_keyword(autism)
    [get_all_drug_evidence(d) for d in x]
    [get_all_associated_targets(d) for d in x]
    [get_disease_disease_associations(d) for d in x]
    print([get_labels_from_evidence(d) for d in x])
```

*Listing 8.* Here we illustrate combinig the PyTDC PrimeKG API with the networkx module to retrieve drug repositioning opportunities.

```python
import networkx as nx
from tdc_ml.resource import PrimeKG

# Load the PrimeKG data
kg = PrimeKG()
data = kg.get_data()
data = data[data[relation].str.contains(drug)]

# Create a graph from the knowledge graph data
G = nx.from_pandas_edgelist(data, 'x_id', 'y_name', edge_attr='relation')

# Example function to find repositioning opportunities for a given drug
def find_repositioning_opportunities(drug):
    neighbors = list(G.neighbors(drug))
    diseases = [node for node in neighbors if G[drug][node]['relation'] == '
        ↪ drug_protein']
    return diseases

# Find repositioning opportunities for a specific drug
drug_name = 'DB00945'
repositioning_opportunities = find_repositioning_opportunities(drug_name)
```

### B.3.3. PYTDC MODEL HUB

The introduced model hub is composed of the PyTDC Model Hub and a set of utilities and endpoints for facilitating model inference and fine-tuning. PyTDC introduces The Commons' HuggingFace Model Hub. It is a resource with pre-trained models, including geometric deep learning models, large language models, and other contextualized multimodal models for therapeutic tasks. The models can be fine-tuned using datasets in PyTDC and be used for downstream tasks such as implementations of multi-agent collaborative schemes (Gao et al., 2024) (i.e., expert consultants) our predictive therapeutic tasks (Theodoris et al., 2023; Initiative, 2023). The model hub details and available models can be found at https://huggingface.co/tdc.

*Listing 9.* The below illustrates the basic functionality of the model hub to download a model and perform inference on a precdictive task as well as fine-tune the model

```
from tdc_ml import tdc_hf_interface
tdc_hf = tdc_hf_interface(BBB_Martins-AttentiveFP)
# load deeppurpose model from this repo
dp_model = tdc_hf.load_deeppurpose('./data')
tdc_hf.predict_deeppurpose(dp_model, ['YOUR_SMILES_STRING'])
# fine-tune
dp_model.train(train, val, test) # for some defined splits
```

*Listing 10.* The below illustrates using the tdc model hub to download a foundation model (Theodoris et al., 2023)

```
from tdc_ml import tdc_hf_interface
from transformers import BertModel
geneformer = tdc_hf_interface(Geneformer)
model = geneformer.load()
assert isinstance(model, BertModel), type(model)
```

*Listing 11.* Beyond downloading a foundation model (Theodoris et al., 2023), the model server facilitates model inference across a range of datasets. Below an example integrating the PyTDC CellXGene API with the model server.

```
from tdc_ml.resource import cellxgene_census
from tdc_ml.model_server.tokenizers.geneformer import GeneformerTokenizer
from tdc import tdc_hf_interface
import torch

# query the CELLXGENE census
adata = self.resource.get_anndata(
var_value_filter=
feature_id_in_['ENSG00000161798',_'ENSG00000188229'],
obs_value_filter=
sex_==_'female'_and_cell_type_in_['microglial_cell',_'neuron'],
column_names={
   obs: [
      assay, cell_type, tissue, tissue_general,
      suspension_type, disease
   ]
},
)

# tokenize gene expression vectors
tokenizer = GeneformerTokenizer()
x = tokenizer.tokenize_cell_vectors(adata,
                     ensembl_id=feature_id,
```

```
                            ncounts=n_measured_vars)

cells, _ = x

# load the model
geneformer = tdc_hf_interface(Geneformer)
model = geneformer.load()

Custom_pre-processing_code_can_include_padding_and_attention_mask_definitions.

input_tensor = torch.tensor(cells)
out = []
for batch in input_tensor:
    # build an attention mask
    attention_mask = torch.tensor(
        [[x[0] != 0, x[1] != 0] for x in batch])
    # run batched inference
    out.append(model(batch, attention_mask=attention_mask))
```

## C. PyTDC model server: models and code

**We have released open source model retrieval and deployment software that streamlines AI inferencing, downstream fine-tuning, and domain-specific evaluation for representation learning models across biomedical modalities.** The PyTDC model server (section 3.2) facilitates the use and evaluation of biomedical foundation models for downstream therapeutic tasks and enforces alignment between new method development and therapeutic objectives. We have released support for inference on several state-of-the-art (SoTA) models (Lopez et al., 2018; Theodoris et al., 2023; Cui et al., 2024; Li et al., 2024; ESM Team, 2024) as well as benchmarking software for domain-specific evaluation (Velez-Arce et al., 2024).

PyTDC presents open source model retrieval and deployment software that streamlines AI inferencing and exposes state-of-the-art, research-ready models and training setups for biomedical representation learning models across modalities. In section 3.2 and figure 2, we describe the architectural components of the model server and its integration into the model development lifecycle with the TDC platform. In figure 1, we show the components of PyTDC enabling the full development lifecycle for transfer learning ML methods in therapeutics. Figure 3 also shares how a workflow spanning hundreds-to-thousands of lines of code is simplified to below 30 with PyTDC's model server and data retrieval capabilities.

We present PyTDC, a machine-learning platform providing streamlined training, evaluation, and inference software for single-cell biological foundation models to accelerate research in transfer learning method development in therapeutics (Theodoris, 2024). PyTDC introduces an API-first (Beaulieu et al., 2022) architecture that unifies heterogeneous, continuously updated data sources. The platform introduces a model server, which provides unified access to model weights across distributed repositories and standardized inference endpoints. The model server accelerates research workflows (Kahn et al., 2022) by exposing state-of-the-art, research-ready models and training setups for biomedical representation learning models across modalities. Building upon Therapeutic Data Commons (Huang et al., 2021; 2022; Velez-Arce et al., 2024), we present single-cell therapeutics tasks, datasets, and benchmarks for model development and evaluation.

Below the set of model server models.

### C.1. scGPT

scGPT (Cui et al., 2024) is A foundation model for single-cell biology based on a generative pre trained transformer across a repository of over 33 million cells.

**Model page.** https://huggingface.co/apliko/scGPT

**Abstract.** Generative pretrained models have achieved remarkable success in various domains such as language and

computer vision. Specifically, the combination of large-scale diverse datasets and pretrained transformers has emerged as a promising approach for developing foundation models. Drawing parallels between language and cellular biology (in which texts comprise words; similarly, cells are defined by genes), our study probes the applicability of foundation models to advance cellular biology and genetic research. Using burgeoning single-cell sequencing data, we have constructed a foundation model for single-cell biology, scGPT, based on a generative pretrained transformer across a repository of over 33 million cells. Our findings illustrate that scGPT effectively distills critical biological insights concerning genes and cells. Through further adaptation of transfer learning, scGPT can be optimized to achieve superior performance across diverse downstream applications. This includes tasks such as cell type annotation, multi-batch integration, multi-omic integration, perturbation response prediction and gene network inference.

**PyTDC Code.** See https://huggingface.co/apliko/scGPT. Illustrated in figure 4.

*Figure 4.* The below code illustrates a workflow using PyTDC's implementation of (Cui et al., 2024). Such a workflow would take a user hundreds-to-thousands of lines of code to develop. PyTDC allows the user to extract single-cell foundation model embeddings from complex and customized gene expression datasets with less than 30 lines of code. See https://huggingface.co/apliko/scGPT

```python
from tdc_ml.multi_pred.anndata_dataset import DataLoader
from tdc_ml import tdc_hf_interface
from tdc_ml.model_server.tokenizers.scgpt import scGPTTokenizer
import torch

# an example dataset
adata = DataLoader("cellxgene_sample_small",
                   "./data",
                   dataset_names=["cellxgene_sample_small"],
                   no_convert=True).adata

# code for loading the model and performing inference
scgpt = tdc_hf_interface("scGPT")
model = scgpt.load()  # This line can cause segmentation fault on inappropriate setup
tokenizer = scGPTTokenizer()
gene_ids = adata.var["feature_name"].to_numpy(
)  # Convert to numpy array
tokenized_data = tokenizer.tokenize_cell_vectors(
    adata.X.toarray(), gene_ids)
mask = torch.tensor([x != 0 for x in tokenized_data[0][1]],
                    dtype=torch.bool)

# Extract first embedding
first_embed = model(tokenized_data[0][0],
                    tokenized_data[0][1],
                    attention_mask=mask)
```

## C.2. Geneformer

Geneformer (Theodoris et al., 2023) is a foundational transformer model pretrained on a large-scale corpus of single cell transcriptomes to enable context-aware predictions in settings with limited data in network biology.

**Model page.** https://huggingface.co/apliko/Geneformer

**Abstract.** Mapping gene networks requires large amounts of transcriptomic data to learn the connections between genes, which impedes discoveries in settings with limited data, including rare diseases and diseases affecting clinically inaccessible tissues. Recently, transfer learning has revolutionized fields such as natural language understanding and computer vision by leveraging deep learning models pretrained on large-scale general datasets that can then be fine-tuned towards a vast array of downstream tasks with limited task-specific data. Here, we developed a context-aware, attention-based deep learning model, Geneformer, pretrained on a large-scale corpus of about 30 million single-cell transcriptomes to enable

context-specific predictions in settings with limited data in network biology. During pretraining, Geneformer gained a fundamental understanding of network dynamics, encoding network hierarchy in the attention weights of the model in a completely self-supervised manner. Fine-tuning towards a diverse panel of downstream tasks relevant to chromatin and network dynamics using limited task-specific data demonstrated that Geneformer consistently boosted predictive accuracy. Applied to disease modelling with limited patient data, Geneformer identified candidate therapeutic targets for cardiomyopathy. Overall, Geneformer represents a pretrained deep learning model from which fine-tuning towards a broad range of downstream applications can be pursued to accelerate discovery of key network regulators and candidate therapeutic targets.

**Code.** See https://huggingface.co/apliko/Geneformer. Illustrated in figure 5

*Figure 5.* The below code illustrates a workflow using PyTDC's implementation of (Theodoris et al., 2023). Such a workflow would take a user hundreds-to-thousands of lines of code to develop. PyTDC allows the user to extract single-cell foundation model embeddings from complex and customized gene expression datasets with less than 30 lines of code. See https://huggingface.co/apliko/Geneformer

```python
from tdc_ml.model_server.tokenizers.geneformer import GeneformerTokenizer
from tdc_ml import tdc_hf_interface
import torch
# Retrieve anndata object. Then, tokenize
tokenizer = GeneformerTokenizer()
x = tokenizer.tokenize_cell_vectors(adata,
                                    ensembl_id="feature_id",
                                    ncounts="n_measured_vars")
cells, _ = x
input_tensor = torch.tensor(cells) # note that you may need to pad or perform other custom
    data processing

# retrieve model
geneformer = tdc_hf_interface("Geneformer")
model = geneformer.load()

# run inference
attention_mask = torch.tensor(
    [[x[0] != 0, x[1] != 0] for x in input_tensor]) # here we assume we used 0/False as a
        special padding token
outputs = model(batch,
                attention_mask=attention_mask,
                output_hidden_states=True)
layer_to_quant = quant_layers(model) + (
    -1
)  # Geneformer's second-to-last layer is most generalized
embs_i = outputs.hidden_states[layer_to_quant]
# there are "cls", "cell", and "gene" embeddings. we will only capture "gene", which is
    cell type specific. for "cell", you'd average out across unmasked gene embeddings per
    cell
embs = embs_i
```

### C.3. PINNACLE

PINNACLE (Li et al., 2024), a flexible geometric deep-learning approach that is trained on contextualized protein interaction networks to generate context-PINNACLE protein representations. Leveraging a human multi-organ single-cell transcriptomic atlas, PINNACLE provides 394,760 protein representations split across 156 cell type contexts from 24 tissues and organs.

**Model page.** https://huggingface.co/apliko/PINNACLE

**Abstract.** Understanding protein function and developing molecular therapies require deciphering the cell types in which proteins act as well as the interactions between proteins. However, modeling protein interactions across biological contexts remains challenging for existing algorithms. Here we introduce PINNACLE, a geometric deep learning approach that generates context-aware protein representations. Leveraging a multiorgan single-cell atlas, PINNACLE learns on contextualized protein interaction networks to produce 394,760 protein representations from 156 cell type contexts across 24 tissues. PINNACLE's embedding space reflects cellular and tissue organization, enabling zero-shot retrieval of the tissue hierarchy. Pretrained protein representations can be adapted for downstream tasks: enhancing 3D structure-based representations for resolving immuno-oncological protein interactions, and investigating drugs' effects across cell types. PINNACLE outperforms state-of-the-art models in nominating therapeutic targets for rheumatoid arthritis and inflammatory bowel diseases and pinpoints cell type contexts with higher predictive capability than context-free models. PINNACLE's ability to adjust its outputs on the basis of the context in which it operates paves the way for large-scale context-specific predictions in biology.

**Code.** See https://huggingface.co/apliko/PINNACLE.

*Figure 6.* The below code illustrates a workflow using PyTDC's implementation of (Li et al., 2024). See https://huggingface.co/apliko/PINNACLE

```
from tdc_ml.resource.pinnacle import PINNACLE
pinnacle = PINNACLE()
embeds = pinnacle.get_embeds() # extract embeddings dataset
```

## C.4. scVI

Single-cell variational inference (scVI) (Lopez et al., 2018) is a powerful tool for the probabilistic analysis of single-cell transcriptomics data. It uses deep generative models to address technical noise and batch effects, providing a robust framework for various downstream analysis tasks. To load the pre-trained model, use the Files and Versions tab files.

**Code.** See https://huggingface.co/apliko/scVI.

*Figure 7.* The below code illustrates a workflow using PyTDC's implementation of (Lopez et al., 2018). See https://huggingface.co/apliko/scVI

```
from tdc_ml.multi_pred.anndata_dataset import DataLoader
from tdc_ml import tdc_hf_interface

adata = DataLoader("scvi_test_dataset",
                   "./data",
                   dataset_names=["scvi_test_dataset"],
                   no_convert=True).adata

scvi = tdc_hf_interface("scVI")
model = scvi.load()
output = model(adata)
```

