# OpenReview forum: "PyTDC: A multimodal machine learning training, evaluation, and inference platform for biomedical foundation models"
_ICML.cc/2025/Conference — ICML 2025 poster_

### Official Review · Reviewer_ckR1 · 2025-03-04

**Overall Recommendation:** 4

**Summary:**

This paper describes PyTDC, an open-source software platform for training, evaluation, and use of biological foundation model. This provides API access to multimodal biological data-sets, a variety of tasks and associated metrics, and APIs for model retrieval and deployment. The utility of this system is demonstrated through the comparison of different models on a context-aware drug target prediction task.

**Claims And Evidence:**

This paper is largely descriptive, rather than making specific scientific claims and providing evidence for them. The stated characteristics and utility of the system is evidenced through high-level diagrams and an example application.

**Essential References Not Discussed:**

The relevant related works are discussed, as far as I can tell

**Experimental Designs Or Analyses:**

The evaluation experiment describe is valid and explained clearly.

**Methods And Evaluation Criteria:**

This work does not present novel methods or evaluation criteria, rather it describes a system for evaluating existing methods under existing evaluation criteria.

**Other Comments Or Suggestions:**

No other comments

**Other Strengths And Weaknesses:**

The paper describes a useful system, which provides a basis for further development and optimisation of multimodal biological foundation models. This should be of general interest to the ICML community and provide a foundation for the development of biological foundation models.

**Questions For Authors:**

No questions

**Relation To Broader Scientific Literature:**

The paper describes a system which looks to be of general use, however I question whether this is the right venue. It would be more suited to a systems-focused workshop, e.g., MLSys.

**Theoretical Claims:**

There are no theoretical claims in this work

---

> ### Author Rebuttal · Authors · 2025-03-26
>
> Reviewer ckR1,
> Thank you very much for the time dedicated to reading the paper and providing your review. Addressing your points below:
> --
> "The paper describes a useful system. The key weakness is a lack of scientific novelty, which makes it more suited to a systems-focused conference or workshop." and "The paper describes a system which looks to be of general use, however I question whether this is the right venue. It would be more suited to a systems-focused workshop, e.g., MLSys."
>
> -> The  ICML 2025 Call for papers lists the following as of interest for the main conference: Machine Learning Systems (improved implementation and scalability, hardware, libraries, distributed methods, etc.). Our paper is inline with this interest.
>
> -> The ICML 2025 Call for papers lists the following as of interest for the main conference: Application-Driven Machine Learning (innovative techniques, problems, and datasets that are of interest to the machine learning community and driven by the needs of end-users in applications such as healthcare, physical sciences, biosciences, social sciences, sustainability and climate, etc.). Our paper is inline with this interest.
>
> -> The paper establishes novelty on the fronts of model evaluation criterion, datasets, task definitions, and machine learning system design.
>
> -> As an example of precedent for ICML publication, our paper cites the ICML paper (Kahn et al., 2022) . It is also an ML systems paper and is a good example we'll cite for some of your other expressed concerns : https://proceedings.mlr.press/v162/kahn22a/kahn22a.pdf
>
> --
>
> "This paper is largely descriptive, rather than making specific scientific claims and providing evidence for them. The stated characteristics and utility of the system is evidenced through high-level diagrams and an example application."
>
> -> This is inline with MLSys publications at ICML such as (Kahn et al., 2022) https://proceedings.mlr.press/v162/kahn22a/kahn22a.pdf
>
> -> Papers presenting new software (models, algorithms, systems, etc) are necessarily descriptive, including those published at ICML. To illustrate this, an ICML paper chosen at random from last year is also largely descriptive with results being showed starting on page 6. https://openreview.net/pdf?id=HsseRq2FAx
>
> --
>
> "This work does not present novel methods or evaluation criteria, rather it describes a system for evaluating existing methods under existing evaluation criteria."
>
> -> The comparisons in Table 1 demonstrate the significant novel contribution of PyTDC.
> -> The novel dataset collection, benchmark organization, metric design, and API architecture are well-executed and demonstrate the platform's utility.
> -> The proposed methods and evaluation criteria seem appropriate for the problem domain. The datasets are well-curated across diverse modalities, and the evaluation metrics appear aligned with both machine learning performance assessment and domain-specific biological relevance, such as out-of-distribution generalizability across cell types.
>
> --
>
> Overall: This review is correct in stating we don't emphasize novel theoretical claims. However, it appears to over-generalize this and argue the work does not present novelty aligned with ICML interests. In this rebuttal, we re-emphasize novelty and how it aligns with ICML Call for papers.
>
> --
>
> In light of this, we kindly request the reviewer re-evaluate their claims and recommendation, and share any further reservations. A revision of the claims made in the review, given inconsistency with ICML criteria and precedent as well as with the content of the paper, and, consequently, the overall recommendation, should be considered.

---

### Official Review · Reviewer_xzws · 2025-03-12

**Overall Recommendation:** 4

**Summary:**

This paper introduces PyTDC, an open-source machine learning platform designed to support the training, evaluation, and inference of multimodal biological AI models, with a strong focus on the single-cell domain. PyTDC facilitates integration of diverse biological data sources from single-cell gene expression, perturbation responses, protein-peptide interactions, and clinical trial data into contextualized machine learning tasks. Key tasks include single-cell drug-target nomination, perturbation response prediction, and cell-type-specific protein-peptide interaction prediction.

A central contribution is the adoption of an API-first architecture, implemented using a Model-View-Controller (MVC) design pattern. This architectural choice allows seamless integration of heterogeneous and continually updated data sources.

The paper also introduces a benchmarking and model retrieval infrastructure, which allows researchers to evaluate state-of-the-art models, fine-tune them on task-specific data, and assess performance on out-of-distribution samples. Notably, the paper highlights the poor performance of existing graph-based and domain-specific methods on the newly introduced tasks, particularly in out-of-distribution contexts, underscoring the need for context-aware, multimodal foundation models in the biomedical domain.

**Claims And Evidence:**

The paper provides reasonably convincing evidence for the overall utility of PyTDC as a multimodal biomedical benchmarking platform. This is supported by a comprehensive case study on the single-cell drug-target nomination task, where several state-of-the-art models (including graph representation learning methods and domain-specific approaches) are benchmarked against the newly introduced contextualized tasks. The case study effectively highlights gaps in current model performance, particularly in out-of-distribution settings, supporting the need for improved multimodal, context-aware methods. The webserver and model retrieval infrastructure appear to be well-implemented and accessible.

However, the necessity for adopting the specific 'API-first' architecture and Model-View-Controller (MVC) design pattern are underexplained. While these approaches are well-established in software engineering, the paper does not clearly justify why they are uniquely advantageous for this use case compared to alternative architectures, including comparison with the previous versions. Further explanation is needed to clarify how this design improves the implementation of the pyTDC server, in perspectives including modularity, scalability, or reproducibility.

**Essential References Not Discussed:**

None to be addressed.

**Experimental Designs Or Analyses:**

The experimental design and case study focused on the single-cell drug-target nomination task were reviewed in detail, including inspection of the provided code examples for loading datasets, training workflows, and model benchmarking results. The experiments are generally sound and well-documented, with clear pipelines for data processing, training, and evaluation.

**Methods And Evaluation Criteria:**

The proposed methods and evaluation criteria seem appropriate for the problem domain. The datasets are well-curated across diverse modalities, and the evaluation metrics appear aligned with both machine learning performance assessment and domain-specific biological relevance, such as out-of-distribution generalizability across cell types.

**Other Comments Or Suggestions:**

Minor comments:

1.	There are several technical issues with references throughout the manuscript, particularly in the Appendix. Compilation errors resulting in `??` citations appear in multiple locations, including: Page 3 line 129 (right column), Page 21, Page 24 lines 1316~1318, Page 38

2.	In Table 1, the check marks and cross marks could benefit from improved visual contrast, such as color-coding or distinct icons, to improve readability. Currently, the marks are difficult to distinguish at a glance.

3.	There is confusion between the terms "pyTDC" and "TDC-2" across the manuscript and appendix. This includes Figure 1 and multiple locations throughout the appendix. The paper needs to be clearer in whether TDC-2 refers to a previous version, a distinct system, or a legacy project in each referenced points for its appearances throughout the manuscript, and ensure terminology is consistent across all sections of the paper.

**Other Strengths And Weaknesses:**

**Strengths**

The overall contribution of this paper is significant, particularly in the context of accelerating multimodal AI research for therapeutic discovery and precision medicine. By combining diverse biological data modalities with a unified benchmarking and inference platform, PyTDC has the potential to serve as a foundation for interdisciplinary studies across computational biology, cheminformatics, and clinical data science.

One of the key strengths is its integration of multimodal data with model retrieval, fine-tuning, and evaluation capabilities, which is often fragmented across different tools and platforms. The open-source nature and availability of a web interface may further lower the barrier to entry for both computational researchers and domain experts.

**Weaknesses**

See `Claims And Evidence` section paragraph 2.

**Questions For Authors:**

1.	While the authors highlight the novelty of implementing the ‘API-first’ architecture via MVC design as one of the main contributions, there are limited demonstration or explanation to support the statement throughout the paper. How does this architecture differ from other versions or databases? In which way might the users benefit from such design? A section for supporting such claim, either quantitatively or qualitatively, may clarify the readers in the technical contribution of pyTDC architecture.

2.	Another area of concern relates to the redistribution and usage of pre-trained models and datasets through the PyTDC platform. Given the diverse licensing terms associated with pre-trained models (some under open-source licenses, others with academic or non-commercial restrictions), how does PyTDC plan to:
    - Clearly surface licensing terms and permissible use when users retrieve models especially through python editors?
    - Ensure that models with restrictive licensing (e.g., non-commercial, academic use only) are not inadvertently used in ways that violate their original licenses?

    Including a transparent licensing and provenance management layer would greatly enhance trust, reproducibility, and adoption in both academic and commercial settings, particularly for users in regulated industries like pharmaceutical research.

**Relation To Broader Scientific Literature:**

The contributions of this paper align with and extend several active research domains at the intersection of multimodal biomedical machine learning, biological data platforms, and therapeutic AI modeling. Specifically, PyTDC expands existing biomedical data platforms by adding multimodal contextualization, foundation model evaluation, and task-specific benchmarking infrastructure, filling a critical gap for researchers developing the next generation of multimodal foundation models for therapeutic discovery.

**Theoretical Claims:**

As I have understood, this manuscript does not present or rely on any formal theoretical claims or mathematical proofs, which is appropriate for a system and benchmarking paper of this nature. The focus is on the development of infrastructure, integration of multimodal data, and empirical benchmarking of biological machine learning tasks.

---

> ### Author Rebuttal · Authors · 2025-03-30
>
> **However, the necessity for adopting the specific 'API-first' architecture and Model-View-Controller (MVC) design pattern are underexplained. While these approaches are well-established in software engineering, the paper does not clearly justify why they are uniquely advantageous for this use case compared to alternative architectures, including comparison with the previous versions. Further explanation is needed to clarify how this design improves the implementation of the pyTDC server, in perspectives including modularity, scalability, or reproducibility.**
>
> We ensure up-to-date retrieval by invoking external APIs rather than storing static data. The chosen MVC design pattern and the resulting “data view” allow us to perform data manipulation per dataset definition and, as such, obtain an ML-ready dataset fitting the task or resource definition. The single-cell gene expression data retrieval API and the shown DSL-based dataset are examples of data views leveraging this pattern, which we can more clearly emphasize in the paper.
>
> **Minor comments:
> There are several technical issues with references throughout the manuscript, particularly in the Appendix. Compilation errors resulting in ?? citations appear in multiple locations, including: Page 3 line 129 (right column), Page 21, Page 24 lines 1316~1318, Page 38
> ...**
>
> Thank you. We will address all of the minor comments raised in the camera-ready version of the paper.
>
> **While the authors highlight the novelty of implementing the ‘API-first’ architecture via MVC design as one of the main contributions, there are limited demonstration or explanation to support the statement throughout the paper. How does this architecture differ from other versions or databases? In which way might the users benefit from such design? A section for supporting such claim, either quantitatively or qualitatively, may clarify the readers in the technical contribution of pyTDC architecture.**
>
> We contrast this architecture difference with other benchmarks in B.1, noting that alternatives, including the previous version (which we can note as well), provide access to “file dumps.” We can reword and expand this to more thoroughly distinguish between static datasets provided by alternatives and our “API-first dataset.” The MVC allows us to implement data views (https://www.w3schools.com/mysql/mysql_view.asp), our chosen implementation of the “API-first dataset” abstraction.
>
> In B.1.3 we state “[PyTDC] develops an Application-Embedded Domain-Specific Data Definition Programming Language facilitating the integration of multiple modalities by generating data views from a mapping of multiple datasets and functions for transformations, integration, and multimodal enhancements, while…”.
>
> We agree that while all components for answering your query are present in the paper, and shared above, a more distinguishable narrative would be helpful and will write an addition to our discussion of the architecture more thoroughly detailing: The API-first architecture and the benefits of invoking external APIs, the “API-first dataset” abstraction, the choice of MVC design pattern, how that implements data views and the overarching “API-first dataset”, and illustrating via existing data views, would be helpful. We cite works demonstrating individual benefits and our contribution is composing these into an innovative design.
>
> **Another area of concern relates to the redistribution and usage of pre-trained models and datasets through the PyTDC platform. Given the diverse licensing terms associated with pre-trained models (some under open-source licenses, others with academic or non-commercial restrictions), how does PyTDC plan to: - Clearly surface licensing terms and permissible use when users retrieve models especially through python editors? -  Ensure that models with restrictive licensing (e.g., non-commercial, academic use only) are not inadvertently used in ways that violate their original licenses? - Including a transparent licensing and provenance management layer would greatly enhance trust, reproducibility, and adoption in both academic and commercial settings, particularly for users in regulated industries like pharmaceutical research.**
>
> We surface license terms on the website for all datasets and will do so for models on our model hub pages as well. We can explore the possibility of augmenting classes to expose licenses more easily. Beyond exposing licenses, it is likely impractical for PyTDC to prevent license violations in its current state as a PyPI package (and Harvard dataverse repository). However, deploying a transparent licensing and provenance layer is certainly a requirement of interest for deploying a distributed service based on the current PyTDC platform. This has been added to our product backlog for future releases.

---

### Official Review · Reviewer_exTK · 2025-03-13

**Overall Recommendation:** 3

**Summary:**

The paper introduces **PyTDC**, a multimodal machine learning platform designed for training, evaluation, and inference of biomedical foundation models. The platform aims to address the limitations of existing biomedical benchmarks by providing end-to-end infrastructure for integrating multimodal biological data and supporting a broad range of machine learning tasks in therapeutics. Key contributions of PyTDC include: (1) integration of single-cell analysis with multimodal machine learning, (2) continuous data updates and heterogeneous data sources, (3) model server for inference and fine-tuning, (4) case study on single-cell drug-target nomination, and (5) context-specific metrics. Overall, PyTDC aims to accelerate research in biomedical AI by providing a unified platform for multimodal data integration, model training, and evaluation, with a focus on therapeutic applications. The platform is open-source and designed to facilitate the development of context-aware, multimodal foundation models for biomedical research.

## Update after Rebuttal
I agree to accept this paper.

**Claims And Evidence:**

As an infrastructure paper, the claims of contributions of PyTDC are concrete. The only concern for me is that the target of PyTDC is supporting "biomedical foundation models", which may be somewhat overclaiming. Biomedicine encompasses a vast range of disciplines, for which various tasks, datasets, and benchmarks have been proposed. There have also been many models claimed to be "biomedical foundation models", which actually focus on very different applications, such as clinical NLP, medical image processing, and drug discovery. Therefore, as an AI platform, PyTDC can hardly cover the whole field of "biomedical foundation models".

**Essential References Not Discussed:**

For the structure-based drug design component of PyTDC, beyond the scoring functions already included, there are two recent popular aspects to be considered in evaluating the molecules:

1. The physical plausibility of 3D structures, such as PoseCheck [1].
2. The diversity of generated molecules, new metrics including NCircles [2] and HamDiv [3].

The SBDD component may not be the main contribution of PyTDC, but as a comprehensive benchmarking platform, I believe these metrics would help.

[1] Harris C, Didi K, Jamasb A, et al. PoseCheck: Generative Models for 3D Structure-based Drug Design Produce Unrealistic Poses[C]//NeurIPS 2023 Generative AI and Biology (GenBio) Workshop.

[2] Xie Y, Xu Z, Ma J, et al. How Much Space Has Been Explored? Measuring the Chemical Space Covered by Databases and Machine-Generated Molecules[C]//The Eleventh International Conference on Learning Representations.

[3] Hu X, Liu G, Yao Q, et al. Hamiltonian diversity: effectively measuring molecular diversity by shortest Hamiltonian circuits[J]. Journal of Cheminformatics, 2024, 16(1): 94.

**Experimental Designs Or Analyses:**

The only experiments in Section 4.3 are completely newly built and act as a case study rather than the main contribution of this paper.

**Methods And Evaluation Criteria:**

The PyTDC platform offers multiple good features for such kinds of infrastructure, including open-source datasets and plug-and-play APIs. The comparisons in Table 1 demonstrate the significant contribution of PyTDC.

**Other Comments Or Suggestions:**

Tiny improvements in writing could be made:

1. In Section 4.3, "PPI" and "ppi" are mixedly used.
2. Some citations in-line act as components of sentences, where \citet should be used instead of \citep. For example, the last line of page 7.
3. "Table" vs. "table", "Figure" vs. "figure", "Section" vs. "section" are mixedly used in many places, which should be unified.

**Other Strengths And Weaknesses:**

None. See above parts.

**Questions For Authors:**

1. How many single-cell data items are there in the dataset? Is this amount enough for training a robust model?
2. Will the PyTDC API be freely available to academic researchers?

**Relation To Broader Scientific Literature:**

PyTDC reviews related works in detail, and its contribution compared to the existing works is significant.

**Theoretical Claims:**

Not applicable.

---

> ### Author Rebuttal · Authors · 2025-03-30
>
> Dear Reviewer exTK, thank you very much for your time dedicated to reviewing our work. Including responses to your queries below.
>
> **The only concern for me is that the target of PyTDC is supporting "biomedical foundation models", which may be somewhat overclaiming. Biomedicine encompasses a vast range of disciplines, for which various tasks, datasets, and benchmarks have been proposed. There have also been many models claimed to be "biomedical foundation models", which actually focus on very different applications, such as clinical NLP, medical image processing, and drug discovery. Therefore, as an AI platform, PyTDC can hardly cover the whole field of "biomedical foundation models".**
>
> This is true. Supporting biomedical foundation models is an aspirational goal in this case. We focus this release on single-cell foundation models and are working to add ESM3 to incorporate the protein sequence, structure, and function modalities. We can either make a stronger note of this limitation or rephrase our claim to be “foundation models in therapeutic discovery”.
>
> **For the structure-based drug design component of PyTDC, beyond the scoring functions already included, there are two recent popular aspects to be considered in evaluating the molecules:
> The physical plausibility of 3D structures, such as PoseCheck [1].
> The diversity of generated molecules, new metrics including NCircles [2] and HamDiv [3].
> The SBDD component may not be the main contribution of PyTDC, but as a comprehensive benchmarking platform, I believe these metrics would help.**
>
> We agree. We are still working on the final release of the codebase, which could be released simultaneously to a camera-ready version. This suggestion has been added to the list of tasks for that release.
>
> **Tiny improvements in writing could be made:
> In Section 4.3, "PPI" and "ppi" are mixedly used.
> Some citations in-line act as components of sentences, where \citet should be used instead of \citep. For example, the last line of page 7.
> "Table" vs. "table", "Figure" vs. "figure", "Section" vs. "section" are mixedly used in many places, which should be unified.**
>
> This will be done for any camera-ready version.
>
> **How many single-cell data items are there in the dataset? Is this amount enough for training a robust model?**
>
> Our website and appendix show the total number of items for all datasets introduced in the paper. For the case study on the single-cell drug target nomination task, we have 394760 (cell, gene) pairs in the dataset. We will mention this explicitly when we release the documentation for the pinnacle resource. This dataset was large enough to train PINNACLE and Node2Vec to satisfactory results. It is possible a larger dataset would have yielded better performance from the tested transformer models. However, we note the dataset presents what is realistically feasible to curate for these diseases based on the current literature.

---

### Official Review · Reviewer_ATdV · 2025-03-14

**Overall Recommendation:** 4

**Summary:**

PyTDC is introduced as a cutting-edge multimodal machine learning infrastructure designed to streamline the training, evaluation, and inference of biomedical foundation models. By unifying heterogeneous, continuously updated data sources and providing a model server for seamless access to pre-trained models and inference endpoints, PyTDC addresses the fragmentation in existing biomedical benchmarks. It supports a wide range of therapeutic tasks, including single-cell drug-target nomination, perturbation response prediction, and protein-peptide interaction prediction, while introducing context-specific metrics to ensure model performance aligns with biomedical goals. As an open-source platform, PyTDC accelerates research by offering modular, customizable tools for multimodal data integration, enabling the development of robust, context-aware foundation models for biomedical AI.

## Update after rebuttal
My Overall Recommendation remains the same.

**Claims And Evidence:**

All the claims in the paper are sound with corresponding contributions in the platform. However, there are two aspects to be clarified:

- PyTDC doesn't include text data, yet natural language is a key modality in biomedicine. Is it appropriate to claim comprehensive support for biomedical foundation models without this modality?
- As a non-commercial platform, how does the group implement the continuous update of new data?

**Essential References Not Discussed:**

None.

**Experimental Designs Or Analyses:**

This paper is not based on comparative experiments and analyses. As an open-source, API-based platform, the user experience of PyTDC will require more feedback from future users to fully evaluate its effectiveness and usability.

**Methods And Evaluation Criteria:**

The platform is highly meaningful, providing essential support for the development of biomedical foundation models. The dataset collection, benchmark organization, metric design, and API architecture are well-executed and demonstrate the platform's utility.

**Other Comments Or Suggestions:**

None.

**Other Strengths And Weaknesses:**

The contribution of PyTDC is groundbreaking in this field, offering a comprehensive and modular solution for biomedical AI research.

**Questions For Authors:**

- What managing mechanism is adopted for continuous data updates in PyTDC?
- How does PyTDC manage the use of data with various licenses? For example, if a foundation model is trained using data accessed through PyTDC from sources with different licenses, how does the platform supervise and ensure compliance with these licenses?

**Relation To Broader Scientific Literature:**

PyTDC provides a fundamental toolkit for the development of biomedical foundation models, addressing a critical gap in the field. The platform's design and capabilities are well-aligned with the needs of modern biomedical AI research.

**Theoretical Claims:**

None.

---

> ### Author Rebuttal · Authors · 2025-03-30
>
> Dear Reviewer ATdV,
> Thank you very much for your time dedicated to reviewing our work. Including our responses to your queries below.
>
> **PyTDC doesn't include text data, yet natural language is a key modality in biomedicine. Is it appropriate to claim comprehensive support for biomedical foundation models without this modality?**
>
> This is fair. Supporting biomedical foundation models is an aspirational goal in this case. We focus this release on single-cell foundation models and are working to add ESM3 to incorporate the protein sequence modality. We could more strongly emphasize this point, or adjust our claim/headline to match “foundation models in therapeutic discovery”.
>
> **As a non-commercial platform, how does the group implement the continuous update of new data?**
>
> This is addressed via an API-first approach. By invoking external APIs rather than storing static data, we assert up-to-date retrieval. The chosen MVC design pattern and the resulting “data view” allow us to perform data manipulation per dataset definition and, as such, obtain an ML-ready dataset fitting the task or resource definition.
>
> **As an open-source, API-based platform, the user experience of PyTDC will require more feedback from future users to fully evaluate its effectiveness and usability.**
>
> This is true. It is hard to fully and rigorously assess the success of this release. However, we have some very positive indicators, including a sharp increase in Github stars (now 1000+), a 2x increase in the Pypi package's MAU, and ~5k invocations of the model server APIs in the last month alone.
>
>
> **What managing mechanism is adopted for continuous data updates in PyTDC?**
>
> Our platform emphasizes an API-first approach rather than storing static data, thus ensuring up-to-date retrieval. However, for static datasets, we cannot yet guarantee continuous data updates.
>
> **How does PyTDC manage the use of data with various licenses? For example, if a foundation model is trained using data accessed through PyTDC from sources with different licenses, how does the platform supervise and ensure compliance with these licenses?**
>
> We don’t yet have rigorous licensing enforcement. However, we expose all licenses for datasets on our website and urge users to review permits for the datasets they are using.

---

### Decision · Program_Chairs · 2025-05-01

**Decision:**

Accept (poster)

**Comment:**

The paper introduces PyTDC, an open-source platform for training, evaluating, and deploying multimodal biomedical machine learning models. It integrates diverse biological data sources, supports a wide range of therapeutic tasks, and provides infrastructure for benchmarking and inference. Reviewers agree that the work addresses a timely and important need in the biomedical AI community. While the contributions are primarily system-oriented, the paper introduces several useful components, including task definitions, evaluation metrics, and a case study that illustrates practical challenges. Some concerns are noted regarding the scope of the claims and architectural clarity, while the overall value of the platform is recognized, especially in facilitating future research in multimodal and context-aware biomedical modeling.